# Drivers of seasonal hydrography in Disko Bay, Greenland

Linda Latuta[1,2], Lars H. Smedsrud[1,2], Elin Darelius[1,2], Per Juel Hansen[3], and Josh K. Willis[4]

[1]Geophysical Institute, University of Bergen, Bergen, Norway
[2]Bjerknes Centre for Climate Research, Bergen, Norway
[3]Department of Biology, Marine Biological Station, University of Copenhagen, Helsingør, Denmark
[4]Jet Propulsion Laboratory, California Institute of Technology, Pasadena, CA, USA

**Correspondence:** Linda Latuta (linda.latuta@uib.no)

**Abstract.**

This study investigates the seasonal dynamics of Disko Bay (Qeqertarsuup Tunua) in west Greenland. On the eastern side, the bay's hydrography is influenced by ice-ocean interactions and exchange with Ilulissat Icefjord (Kangiata Sullua), while on the western side, the bay exchanges waters with Baffin Bay. Since the mid-1990s, this region has experienced ocean warming, sea-ice decline, and the retreat of Greenland's fastest-flowing marine-terminating glacier. Although West Greenland Irminger Water (WGIW) is known to be a significant heat source behind these changes, the timing and pathways of its entry into Disko Bay remain poorly understood. We present a two-year (2022–2024) observational record of Disko Bay hydrography, providing new insights into the seasonal evolution and spatial structure of Polar Water (PW) and WGIW. Each spring, dense WGIW crosses the topographic barrier between Baffin Bay and Disko Bay, filling the Disko Bay basin and producing the highest observed temperature and density at depth. The PW–WGIW boundary shoals to depths shallow enough for WGIW to renew the Ilulissat Icefjord basin. In autumn/winter 2022, an additional episodic renewal coincided with strong upwelling-favourable winds along the west Greenland shelf. While WGIW renewal dominates winter and spring seasonality (a period also marked by sea-ice presence), summer and autumn hydrography are shaped by PW. With the onset of the melt season, a fresh stratified layer forms in the upper 50 m and progressively thickens and extends downward, continuing to freshen and cool through autumn. Beneath this layer, denser PW warms steadily along isopycnals, with spatial analyses indicating an advective pathway transporting this warming signal along the bay's periphery.

## 1 Introduction

Disko Bay (Qeqertarsuup Tunua) is the largest open-water embayment in west Greenland (Fig. 1). Bordered by Baffin Bay to the west and numerous glacial fjords to the east, its hydrography is shaped by a complex interplay of regional water mass exchanges, local oceanographic processes, and ice-ocean interactions. Since the late 1960s, oceanographic studies have provided valuable insights into its hydrography and circulation patterns (Petersen, 1964; Muench, 1971; Andersen, 1981a). Subsequent research explored seasonality and ecosystem functioning (Andersen, 1981b; Nielsen TG and Hansen B, 1995), while more recent studies have documented significant changes in oceanographic conditions over the recent decades (Hansen et al., 2012; Myers and Ribergaard, 2013).

One of the most notable changes was a transition in the mid-1990s, when the increased presence of warm Atlantic-origin waters marked the shift from a cold to a warm regime in Disko Bay (Holland et al., 2008; Hansen et al., 2012). This warming was particularly evident at 200-250 m depth, where temperatures rose from $\sim$1.5°C in the early 1990s to $\sim$2-2.5°C by the late 1990s, eventually surpassing 3°C in the early 2000s (Hansen et al., 2012; Khazendar et al., 2019; Joughin et al., 2020).

These warm waters, upon entering Ilulissat Icefjord (Kangiata Sullua) above its 245 m sill (Fig.1b), have contributed to the disintegration of the floating ice tongue, retreat, acceleration, and increased melting of Sermeq Kujalleq (Jakobshavn Glacier), Greenland's fastest flowing marine-terminating glacier (Joughin et al., 2004; Holland et al., 2008; Motyka et al., 2011; Khazendar et al., 2019; Joughin et al., 2020; Picton et al., 2025). More recently, oceanic forcing has also been linked to a period of slowdown, thickening, and terminus re-advance in 2016-2018 (Joughin et al., 2018, 2020), as well as to renewed acceleration and increased solid ice discharge in subsequent years (Picton et al., 2025). Because Sermeq Kujalleq is highly sensitive to oceanic forcing, and the waters that fill Ilulissat Icefjord originate in Disko Bay (Gladish et al., 2015a), an improved understanding and continued monitoring of Disko Bay's hydrography is essential.

Although Disko Bay has been the focus of long-term observations, data coverage is biased toward April—September period. As a result, key processes such as the seasonality of warm subsurface waters remain poorly understood. While it has been hypothesised that these deep warm waters renew in Disko Bay during winter and early spring (Gladish et al., 2015a), the lack of observations during these seasons has left this process largely unexplored.

This study addresses these gaps by presenting oceanographic observations spanning two annual cycles from June 2022 to November 2024. These observations provide new insights into the seasonal processes shaping Disko Bay's hydrography and its response to external forcing, particularly during the poorly observed autumn-to-spring transition

## 2   Regional setting

The regional circulation in Baffin Bay consists of two major currents: the northward-flowing West Greenland Current (WGC) and the southward-flowing Baffin Island Current (BIC) (Fig. 1a). The WGC carries warm and saline subsurface waters of North Atlantic origin along the continental shelfbreak of west Greenland (Pacini et al., 2020; Huang et al., 2024). While most of these warm waters divert west and south in the northern Labrador Sea, some continue northward through Davis Strait into Baffin Bay (Cuny et al., 2005; Curry et al., 2011, 2014). Upon crossing the Davis Strait, the warm waters subduct below fresher and colder Polar-origin waters and propagate northward as a bottom-intensified current along the continental slope (Huang et al., 2024). Despite the heat loss caused by mixing with the Polar-origin waters above, these subsurface warm waters remain the predominant heat source and a major driver of accelerated melt of many marine-terminating glaciers along the west Greenland coast (Holland et al., 2008; Straneo et al., 2012; Khazendar et al., 2019; Joughin et al., 2020; Wood et al., 2021).

The exchange of waters between Baffin Bay and Disko Bay is strongly modulated by topography. A 300–900 m deep trough, Egedesminde Dyb (ED), cuts across the continental shelf and provides a pathway for dense, warm waters from the shelf break into Disko Bay (Fig. 1a). This inflow is partially obstructed by Egedesminde Dyb Sill (EDS), a topographic barrier reaching 300 m depth that functions similarly to a fjord sill (Fig. 1b) (Gladish et al., 2015a). We view EDS as the western

boundary of Disko Bay and the approximate delineation of the Disko Bay–Baffin Bay boundary. The coast defines the eastern and southern boundaries. In the east, Disko Bay connects with Ilulissat Icefjord (750–800 m deep), which is separated by a shallow sill (deepest point of 245 m), also known as the Iceberg Bank (Gladish et al., 2015b; Morlighem et al., 2022). Disko Island is located in the north of the bay, separated from the mainland by the Vaigat Strait (Sullorsuaq Strait), where a shallow bathymetry (245 m) also restricts exchange (Andersen, 1981a; Morlighem et al., 2022). Thus, Disko Bay (100 km wide and $\sim$120 km long) is confined by shallow and complex bathymetry, with its central basin (300–500 m deep) isolated from direct water mass exchanges.

Previous studies indicate a cyclonic circulation within Disko Bay (Andersen, 1981a; Sloth and Buch, 1984), following bathymetric contours, with northward flow past Ilulissat Icefjord (Beaird et al., 2017), and outflow primarily through Vaigat Strait and along the southern coast of Disko Island (Andersen, 1981a; Hansen et al., 2012).

## 3 Data and methods

### 3.1 Oceanographic observations

We combine new observations from two drifting profilers and a hydrographic field campaign with existing data from the monitoring station in Disko Bay to construct a hydrographic time series for 2022–2024 (Fig.2). Table 1 gives an overview of all hydrographic profiles included in the time series, and profile locations are shown in Fig. 1b. Additionally, we utilise profiles from five hydrographic surveys between 2022 and 2024 for a spatial analysis.

#### 3.1.1 Monitoring Station

We use hydrographic data from the Greenland Ecosystem Monitoring Programme (GEM), collected on board RV *Porsild* at a fixed oceanographic monitoring station in northwestern Disko Bay (Table 1, Fig. 1b, Greenland Ecosystem Monitoring (2025a)). We obtained all available processed profiles, taken with a Sea-Bird SBE 19plus instrument from June 2022 to November 2023, and with an AML Oceanographic AML-6 instrument from May to November 2024. Instrument accuracies are given in Table 1.

To improve temporal resolution during the undersampled autumn period, we also conducted weekly measurements at the monitoring station from RV *Porsild* between August and November 2023 (Table 1, Fig. 1b), using the same Sea-Bird SBE 19plus instrument as GEM. We processed the raw data using Sea-Bird Scientific's SBE Data Processing (v7.26.7) application, following standard quality control, correction, and processing steps. In addition, one profile was taken in March 2023 at the landfast sea-ice edge, located 2.8 km landward from the monitoring station (Table 1, Fig. 1b).

We combined our field observations (including the profile taken from sea ice) with GEM data to construct a spatially fixed time series, hereafter referred to as "Monitoring Station" (Fig.2a,c). We use TEOS10 Gibbs-Sea Water Oceanographic Toolbox (McDougall and Barker, 2011) to convert conductivity to Absolute Salinity ($S_A$), temperature to Conservative Temperature

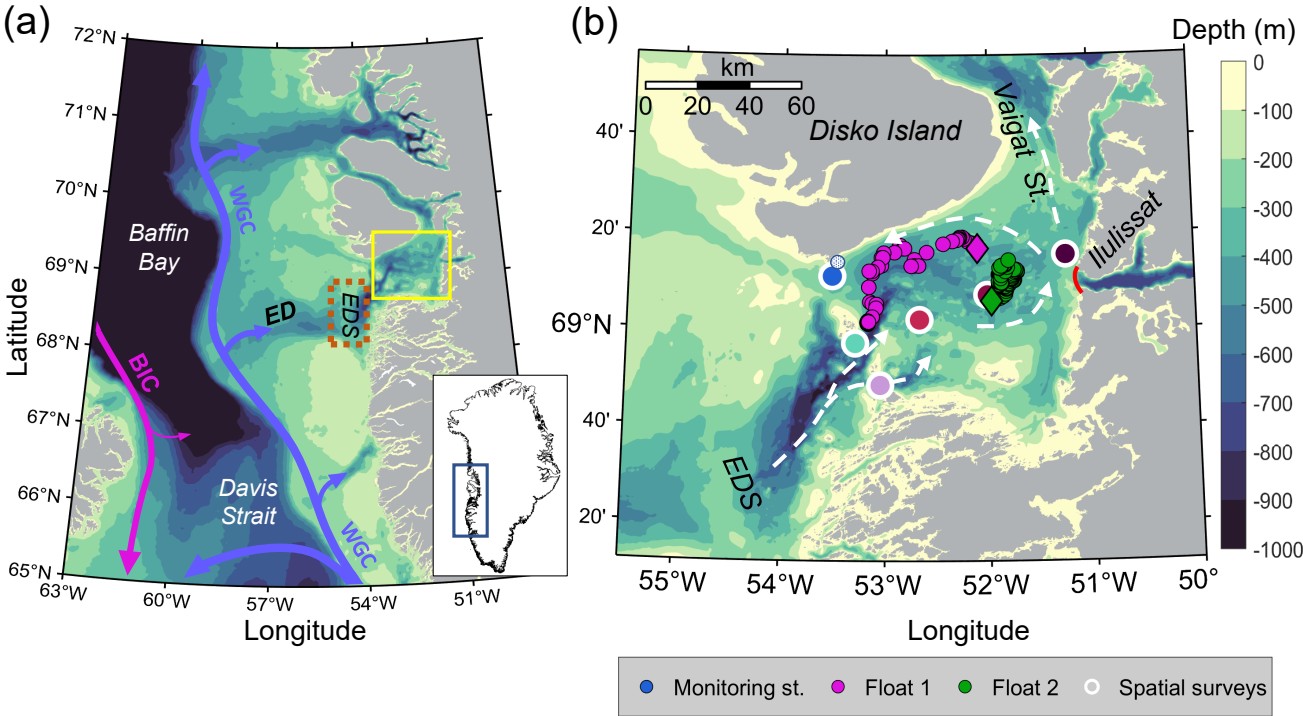

**Figure 1.** (a) Baffin Bay and west Greenland bathymetry, overlaid with north-flowing West Greenland Current (WGC, blue) and south-flowing Baffin Island Current (BIC, pink). The Egedesminde Dyb trough (ED) cuts across the continental shelf into Disko Bay (yellow box). The shallow Egedesminde Dyb Sill (EDS) is where we analyse wind forcing with ERA5 reanalysis data (orange dashed box). (b) Disko Bay bathymetry, Ilulissat Icefjord Sill (red line) and general circulation (white arrows, adapted from Hansen et al. (2012)). Location of oceanographic observations: Monitoring Station (blue circle with white outline), profile from landfast sea-ice (white circle with blue outline), Float 1 (Aug 2022–June 2023, magenta circles), Float 2 (Aug 2023–Oct 2024, green circles), and stations surveyed during spatial cruises (coloured circles with thick while outline, including the Monitoring Station). Each float's first profile is marked with a diamond. Bathymetric data are from BedMachine Version 5 dataset (Morlighem et al., 2022).

($\Theta$), and pressure to depth. Throughout this paper, we refer to Absolute Salinity and Conservative Temperature as salinity and temperature, respectively. Hereafter, we refer to potential density anomaly with a reference pressure of 0 dbar ($\sigma_0$) as density.

### 3.1.2 Profiling float data

We use temperature and salinity profiles from an Air-Launched Autonomous Micro Observer (Alamo) float (ID: F9313), deployed in Disko Bay as part of the NASA Oceans Melting Greenland (OMG) Mission (Table 1,Oceans Melting Greenland (2022)). The float drifted cyclonically within the bay at a mean speed of 0.4 cm s$^{-1}$, with a higher mean of 0.75 cm s$^{-1}$ while drifting westward before turning south and slowing (Fig. 1b). Between 6 February and 4 April 2023, the float profiled

**Table 1.** Overview of analysed hydrographic observations.

| Name | Period | Location | Profiles | Sampling freq. | Instrument | Operated by |
|---|---|---|---|---|---|---|
| Monitoring St. | Jun 2022–Nov 2024 | 69°10'N 53°31'W | 23 | monthly | SBE 19plus[a] / AML-6[b] | GEM |
| Monitoring St. | Aug–Nov 2023 | 69°10'N 53°31'W | 6 | weekly | SBE 19plus[a] | fieldwork |
| Monitoring St. (sea ice) | Mar 2023 | 69°12'N 53°31'W | 1 | once | SBE 19plus[a] | fieldwork |
| Float 1 | Aug 2022–Jun 2023 | see Fig. 1b (trajectory) | 63 | 5-day (from 22 Sep 2022) | RBR[c] | NASA OMG |
| Float 2 | Aug 2023–Oct 2024 | see Fig. 1b (trajectory) | 74 | 5-day (from 17 Oct 2023) | RBR[c] | GOO |
| Hydrographic surveys | Aug 2022; May & Aug 2023–2024 | Fig. 1b (stations) | 6 per survey | near-synoptic | SBE 19plus[a] / AML-6[b] | GEM |

[a] Sea-Bird SBE 19plus (T: $\pm0.005$ °C; C: $\pm0.005$ mS cm$^{-1}$; P: 0.1% FS).    [b] AML-6 (T: $\pm0.005$ °C; C: $\pm0.01$ mS cm$^{-1}$; P: 0.05% FS).    [c] RBR sensors (T: $\pm0.002$ °C; C: $\pm0.003$ mS cm$^{-1}$; P: $\pm1$ dbar).

*Note:* At the Monitoring Station and in hydrographic surveys, SBE 19plus was used in 2022–2023 and AML-6 in 2024.

underneath the sea ice, and position data during that period were unavailable. However, between the acquisitions with known positions, the float's position changed only by 3.1 km. The float data were quality controlled following the recommended procedures in Wong et al. (2024). Hereafter, we refer to this profiling float as "Float 1" (Fig.2b,d).

     Additional profiles were retrieved by an Apex float (WMO ID: 6990591) (Argo, 2024), deployed in Disko Bay in August 2023 as part of the Greenland Ocean Observations (GOO) project (Table 1, Fig. 1b). The data are Real-time and quality-

controlled. We use only the data with a "good data" quality flag and follow the same quality control checks and processing as for Float 1. Hereafter, we refer to this profiling float as "Float 2" (Fig.2b,d).

     Both floats were fitted with RBR sensors (Table 1). Salinity obtained from both floats was compared against CTD observations collected at the Monitoring Station during periods of temporal overlap, and against data from the hydrographic surveys (Section 3.1.3). These comparisons confirmed that the $\Theta$-$S_A$ relationship at higher densities exhibits spatial heterogeneity

across Disko Bay (see Section 4.2). Accordingly, float salinities in these density ranges were compared against the full set of available CTD observations in $\Theta$-$S_A$ space, from which we estimate that salinity sensor drift did not exceed 0.02 PSU over the period of data used from either float.

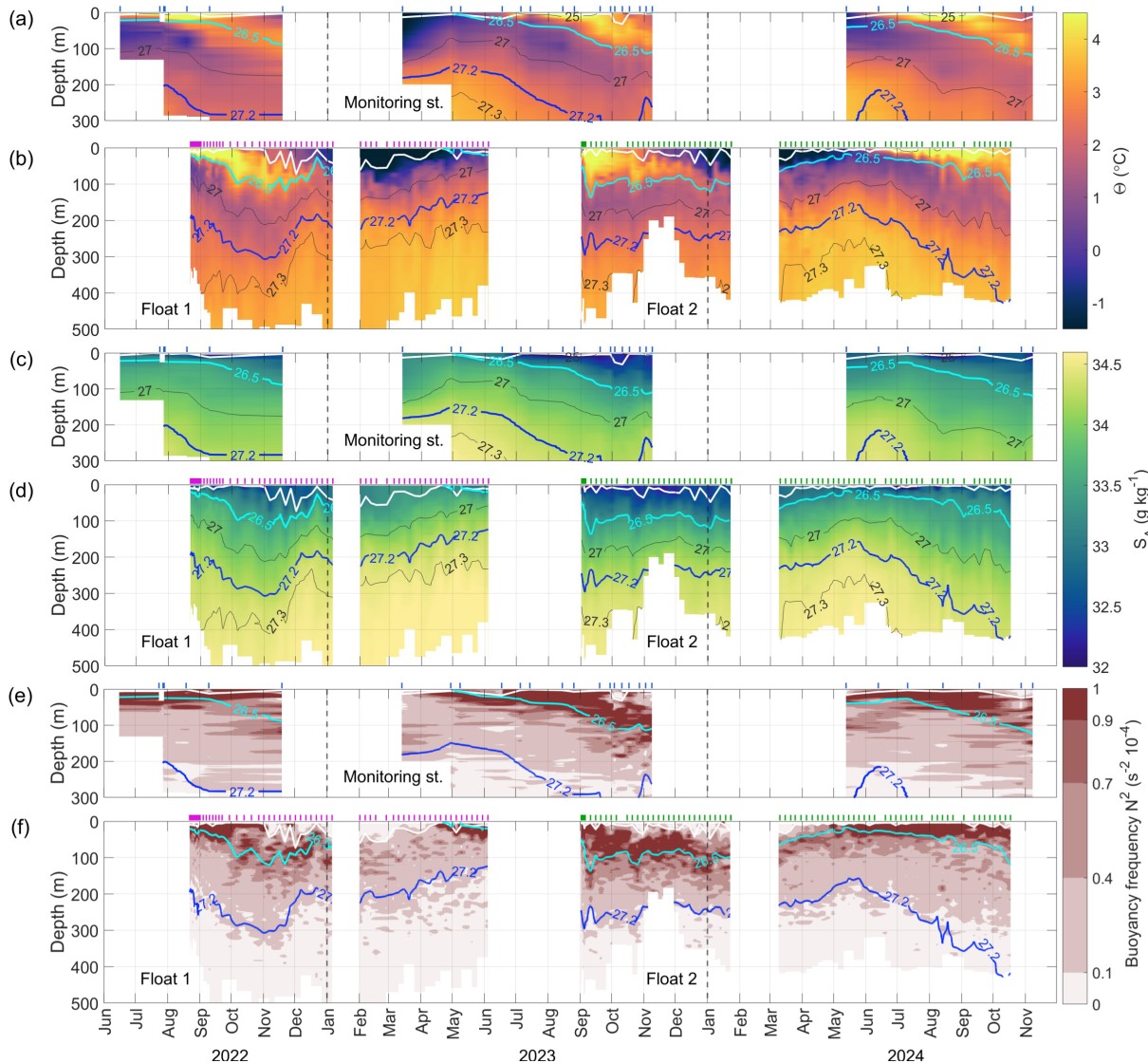

**Figure 2.** Hydrography in Disko Bay from June 2022 to November 2024. Temperature for the Monitoring Station (a) and profiling floats (b). Salinity for the Monitoring Station (c) and profiling floats (d). Buoyancy frequency for the Monitoring Station (e) and profiling floats (f). Temperature, Salinity, and Buoyancy Frequency are overlaid by labelled isopycnals (blue thick line $\sigma_0$= 27.2 kg m$^{-3}$ showing the upper WGIW boundary) and mixed-layer depth (white lines). Vertical ticks on the upper x-axis in panels (a–f) show the time of profile acquisitions (blue –Monitoring Station, magenta – Float 1, green – Float 2), while dashed vertical lines through all panels mark the start of each calendar year.

### 3.1.3 Spatial hydrographic surveys in 2022–2024

We use data from five hydrographic surveys conducted in Disko Bay during the summers of 2022, 2023 and 2024 (Table
1, Fig. 1b) to analyse the spatial variability during this period. These near-synoptic cruises were conducted as part of the
GEM programme (Greenland Ecosystem Monitoring, 2025b), covering most of Disko Bay's deep basin. Instrumentation was
consistent with the Monitoring Station observations: a Sea-Bird SBE 19plus in 2022 and 2023, and an AML Oceanographic
AML-6 in 2024. The cruises took place on 18–21 August 2022, 9–12 May and 14–16 August 2023, as well as 16–19 May and
21–22 August 2024. All cruises had the same sampling locations marked in Fig. 1b.

### 3.2 Atmospheric and sea-ice data


To estimate atmospheric forcing on local ocean variability in Disko Bay and across the shelf region, we use the European
Centre for Medium-Range Weather Forecasts ERA5 reanalysis product (Hersbach et al., 2023). The product has 0.25° spatial
and hourly temporal resolution. We obtained sea-ice concentration and 10-m $u$ and $v$ wind components over a region covering
EDS (Fig.1a).
For sea-ice concentration inside Disko Bay, we use data from the merged MODIS-AMSR2 satellite product (Ludwig et al.,
2020). Daily sea-ice concentration (SIC) with 1km resolution was downloaded for the Disko Bay region (68°42'N – 69°31'N,
51°24'W – 53°W) and averaged spatially to obtain a time series of weekly mean SIC.

### 3.3 Methods

#### 3.3.1 Determination of the mixed-layer depth

We determined the mixed-layer depth (MLD) for each hydrographic profile using the two-step method of Semper et al. (2024),
adapted for use with density profiles. First, for each profile, we obtained the preliminary MLD by computing the normalised
sum-of-squared errors (SSE) over all possible surface-to-depth ranges. Within the mixed layer, normalised SSE values re-
mained small but increased once stratified waters were included as the depth range was extended below the mixed layer.
The preliminary MLD was taken as the maximum depth for which normalised SSE values stayed below the threshold of
$1.5 \times 10^{-4}$ $kg^2$ $m^{-6}$. In some cases, no MLD was detected, either because the mixed layer was shallower than the first avail-
able measurement depth or because no well-mixed surface layer was present.

     Second, we verified the preliminary MLD by checking whether temperature, salinity, and density from the surface to the
computed MLD lay within one standard deviation of their respective mean values (Pickart et al., 2002; Semper et al., 2024). In
a small number of profiles where one or more properties fell outside the standard deviation envelope, the MLD was manually
reassigned to satisfy this criterion.

### 3.3.2 Water mass definitions

Two water masses broadly describe the vertical structure of Disko Bay: a relatively cool and fresh layer of Arctic origin (solid lines in Fig. 3d-e), which overlays warmer and more saline waters of Atlantic origin (dashed lines in Fig. 3d-e).

The Atlantic-origin waters in the region are often called Modified Irminger Water (Gladish et al., 2015a), Subpolar Mode Water (Rysgaard et al., 2020), Atlantic Water (Beaird et al., 2017) and West Greenland Irminger Water (Curry et al., 2014; Carroll et al., 2018; Huang et al., 2024). These definitions broadly overlap, and we refer to this water mass as West Greenland Irminger Water (WGIW) and define it as waters with a density $\sigma_0 > 27.2$ kg m$^{-3}$ (thick grey isopycnal in Fig. 3a-c). This definition matches the properties presented by Curry et al. (2014) and those found to be relevant for Disko Bay and Ilulissat Icefjord basin exchanges (Gladish et al., 2015a, b).

The Arctic-origin waters have also been described using multiple names, depending on origin and formation processes: West Greenland Shelf Water and Arctic Water (Curry et al., 2014; Carroll et al., 2018), Baffin Bay Polar Water and Coastal Water (Rysgaard et al., 2020), cold Polar Water and warm Polar Water (Huang et al., 2024). Because we focus on seasonal evolution rather than source differentiation, we use the general term Polar Water (PW), similar to Myers and Ribergaard (2013), Beaird et al. (2017) and Muilwijk et al. (2022). The PW has density $\sigma_0 < 27.2$ kg m$^{-3}$. Through this paper, we also refer to the surface layer, which is defined as the extent of the mixed layer found within PW. We emphasise that this PW definition merges waters from distinct sources into a single, cool and fresh layer. This simplification is necessary as distinguishing individual water mass sources/types is challenging with the data used in this study. Although each of the water types within our PW definition may have distinct origins and seasonal behaviours, we will assess their combined effect on the upper-layer hydrography in Disko Bay.

Glacial freshwater input, a significant component of what we term PW in Disko Bay, primarily originates from Ilulissat Icefjord. This includes both liquid and solid fluxes, with the liquid component consisting of runoff and Submarine Melt Water (SMW) (Mernild et al., 2015; Enderlin et al., 2016; Beaird et al., 2017). SMW forms through direct melting of marine-terminating glaciers or icebergs by ocean heat ($\Theta = $ -90°C, $S_A = 0$ g kg$^{-1}$) (Gade, 1979). Runoff, from surface melt of glaciers and snow, typically enters the fjord at depth via subglacial pathways, forming Subglacial Discharge (SGD, $\Theta = 0$°C, $S_A = 0$ g kg$^{-1}$)(Straneo and Cenedese, 2015). Together, SGD and SMW drive convective upwelling in the fjord, entraining ambient waters and producing Glacially Modified Water (GMW) (Straneo et al., 2011; Beaird et al., 2015; Stevens et al., 2016; Beaird et al., 2018; Mortensen et al., 2020; Muilwijk et al., 2022). The GMW equilibrates at its level of neutral buoyancy (Jackson et al., 2017; Cowton et al., 2015; Mankoff et al., 2016; Slater et al., 2022) and is exported into Disko Bay if it extends above the Ilulissat Icefjord sill depth (Jenkins, 2011; Gladish et al., 2015b; Carroll et al., 2016; Beaird et al., 2017; Kajanto et al., 2023).

### 3.3.3 Wind stress and Ekman pumping calculation

Zonal wind stress ($\tau_x$) and meridional wind stress ($\tau_y$) are computed for each grid point of the ERA5 fields of $u_{10}$ and $v_{10}$ wind speed components as follows:

$$\tau_x = \rho_a C_d u_{10} U_{10}, \tau_y = \rho_a C_d v_{10} U_{10} \tag{1}$$

where $\rho_a$ is air density (1025 kg m$^{-3}$), $U_{10} = \sqrt{u_{10}^2 + v_{10}^2}$ is the magnitude of the wind vector, and $C_d$ is the drag coefficient. To incorporate the effect of sea ice on the surface wind stress, we use a parameterisation $C_D = C_d(A)$, where A is sea ice concentration (Lüpkes and Birnbaum, 2005). The Ekman pumping velocity ($W_{Ek}$) is then calculated with the curl of the surface wind stress as

$$W_{Ek} = \frac{1}{\rho_0 f_0} \left( \frac{\partial \tau_y}{\partial x} - \frac{\partial \tau_x}{\partial y} \right) \tag{2}$$

where $f_0$ is Coriolis parameter calculated for each latitude, $\rho_0$= 1027.0 kg m$^{-3}$ is a reference density and $\partial x$ and $\partial y$ are grid size.

## 4 Results

### 4.1 Water masses

The two principal water masses are evident in the Temperature–Salinity ($\Theta$–$S_A$) diagrams from the Monitoring Station and Floats (Fig. 3a–c). At the surface, PW varies strongly over the seasonal cycle, with temperatures ranging from the in situ freezing point in winter to $\sim 8°$C in summer (Fig. 3a–c, d) and salinity ranging from $\sim 31$ g kg$^{-1}$ in summer to 33–33.4 g kg$^{-1}$ in winter and early spring (Fig. 3a–c, e).

Between 50 m depth and the WGIW boundary, PW maintains a broad temperature range ($\sim 0$ to 4°C), with the coldest subsurface temperatures in winter and early spring (Fig. 3b–c, d). After sea-ice melt, surface waters re-stratify and warm, while subsurface PW retains a characteristic temperature minimum. This minimum is most pronounced in summer and persists at greater depths in autumn (Fig. 3d).

Mixing with other water sources accounts for the wide property range of PW. Fresh meltwater pulls the near-surface observations along the runoff line in spring (Fig. 3a–c). Similarly, early autumn observations align with the runoff line, while later in autumn they shift toward the SMW mixing line (Fig. 3a–c).

In contrast, WGIW ($\sigma_0 > 27.2$ kg m$^{-3}$) shows less temporal variability, with temperatures of 2–4°$C$ and salinity near $\sim 34.6$ g kg$^{-1}$ (Fig. 3a–e).

### 4.2 Spatial variability

The five near-synoptic GEM cruises (summers 2022–2024), together with overlapping Float 1 and 2 profiles, reveal significant spatial variability in PW properties across Disko Bay (Fig. 4). This variability is especially pronounced in August, with differences of up to 3°C along isopycnals in the $\sigma_0 < 27.1$ kg m$^{-3}$ range (Fig. 4b, d, f). In all August cruises, the coldest upper layer consistently occurred north of Ilulissat Icefjord, while the southwestern region near Aasiaat exhibited a notably warmer subsurface PW relative to all other areas. The spatial contrasts were most distinct in August 2022 (Fig. 4b), when the southern arm of the trough (near Aasiaat) contained the warmest subsurface PW, the middle arm was slightly cooler, and progressively

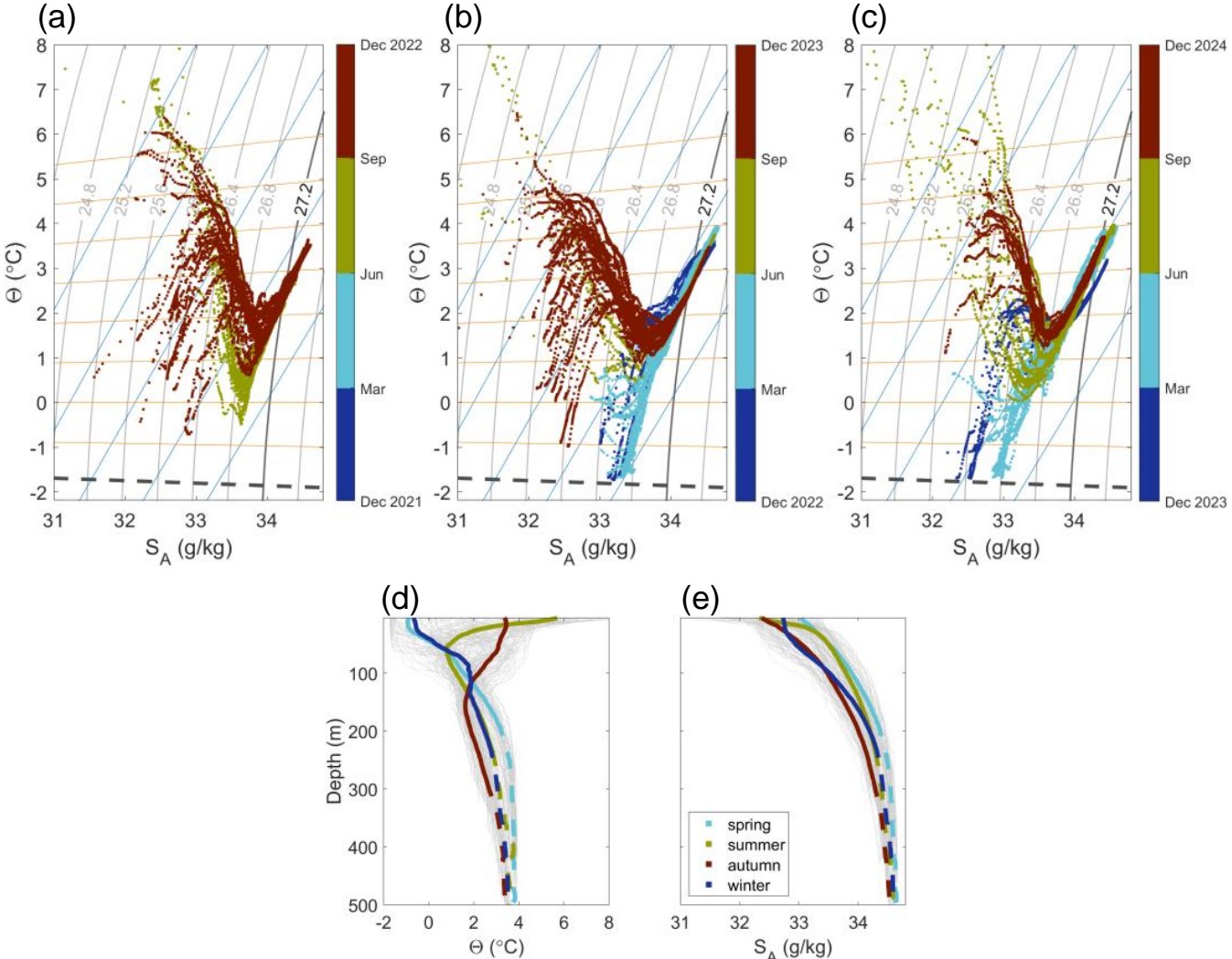

**Figure 3.** (a–c) Temperature–Salinity ($\Theta$–$S_A$) diagrams from Monitoring Station and float observations for annual cycles spanning June 2022 - November 2024 (circle markers coloured by season). Grey contours are isopycnals at 0.4 kg m$^{-3}$ intervals; the thick grey line at 27.2 kg m$^{-3}$ delineates West Greenland Irminger Water (WGIW) from Polar Water (PW). Orange and blue lines indicate mixing lines with subglacial discharge/runoff ($\Theta = 0°$C, $S_A = 0$ g kg$^{-1}$) and submarine meltwater ($\Theta = -90°$C, $S_A = 0$ g kg$^{-1}$), respectively. The dashed grey line marks the surface freezing point. (d–e) Vertical temperature and salinity profiles from all observations in (a–c) (thin grey lines), overlaid with mean seasonal profiles (thick coloured lines, with solid segments corresponding to PW and dashed segments to WGIW).

colder conditions were observed toward the station nearest to the trough, then north of Ilulissat Icefjord, and finally at the Monitoring Station south of Disko Island. The central basin was the coldest area overall, aside from a sharp subsurface temperature
minimum at the Monitoring Station.

**Table 2.** Spatial variability of West Greenland Irminger Water (WGIW) across Disko Bay from GEM spatial surveys (2022–2024). Variability is reported as the standard deviation of PW–WGIW boundary depth and density at 300 m and 400 m depth. Values are calculated within each cruise across all stations.

| Year | Mean PW–WGIW boundary depth (m) | SD of PW–WGIW boundary depth (m) | $\sigma_0$ SD at 300 m (kg m$^{-3}$) | $\sigma_0$ SD at 400 m (kg m$^{-3}$) |
|---|---|---|---|---|
| 2022 Aug | 243 | 25 | 0.0187 | 0.0132 |
| 2023 May | 157 | 29 | 0.0156 | 0.0148 |
| 2023 Aug | 239 | 24 | 0.0201 | 0.0048 |
| 2024 May | 232 | 19 | 0.0262 | 0.0175 |
| 2024 Aug | 320 | 23 | 0.0343 | 0.0217 |

In addition to along-isopycnal variability, PW also exhibits substantial variability along depth levels, particularly in August. The pattern is consistent with isopycnals sloping downward toward the coast, as previously documented north of Ilulissat Icefjord (Beaird et al., 2017). In the spatial cruises, the same tendency is evident at the near-coastal stations both north of Ilulissat Icefjord and near Aasiaat, where, for example, the $\sigma_0$ = 26.7 kg m$^{-3}$ isopycnal was more than 60 m deeper than in the central parts of the bay.

In contrast, WGIW exhibits much weaker spatial variability. Near-synoptic observations within WGIW fall along a narrow line in the $\Theta$–$S_A$ space (Fig. 4b–f), indicating minimal variability along isopycnals. Differences are somewhat greater along depth levels, but remain small: the PW–WGIW boundary depth varied by 20–30 m across the bay, and standard deviations in density at fixed depths were relatively small and decreased with depth. The average within-cruise standard deviation was ∼0.023 kg m$^{-3}$ at 300 m and ∼0.014 kg m$^{-3}$ at 400 m, with slightly larger variability in 2024 (both May and August), compared to 2022 and 2023 (Table 2).

Because floats sampled the deeper parts of the bay, analysis of WGIW seasonality relies largely on float data. These observations can be used to assess WGIW variability along isopycnals with confidence, as well as at depth levels, provided that the variability in time exceeds the background spatial variability noted above. The wide spatial coverage of Float 1 in 2022–2023 is considered acceptable for studying WGIW properties, as that period showed limited spatial variability (Fig. 4b, c; Table 2). In contrast, PW seasonality is better assessed from fixed-point observations at the Monitoring Station, since the pronounced spatial variability in PW would obscure the seasonal patterns if Float data are used.

### 4.3 Surface mixed-layer modifications

The MLD, mixed-layer salinity, and mixed-layer temperature all exhibit a consistent annual cycle over the observation period (Fig. 5). During winter, mixed-layer temperatures remain at the in situ freezing point (Fig. 5b), while sea-ice formation in-

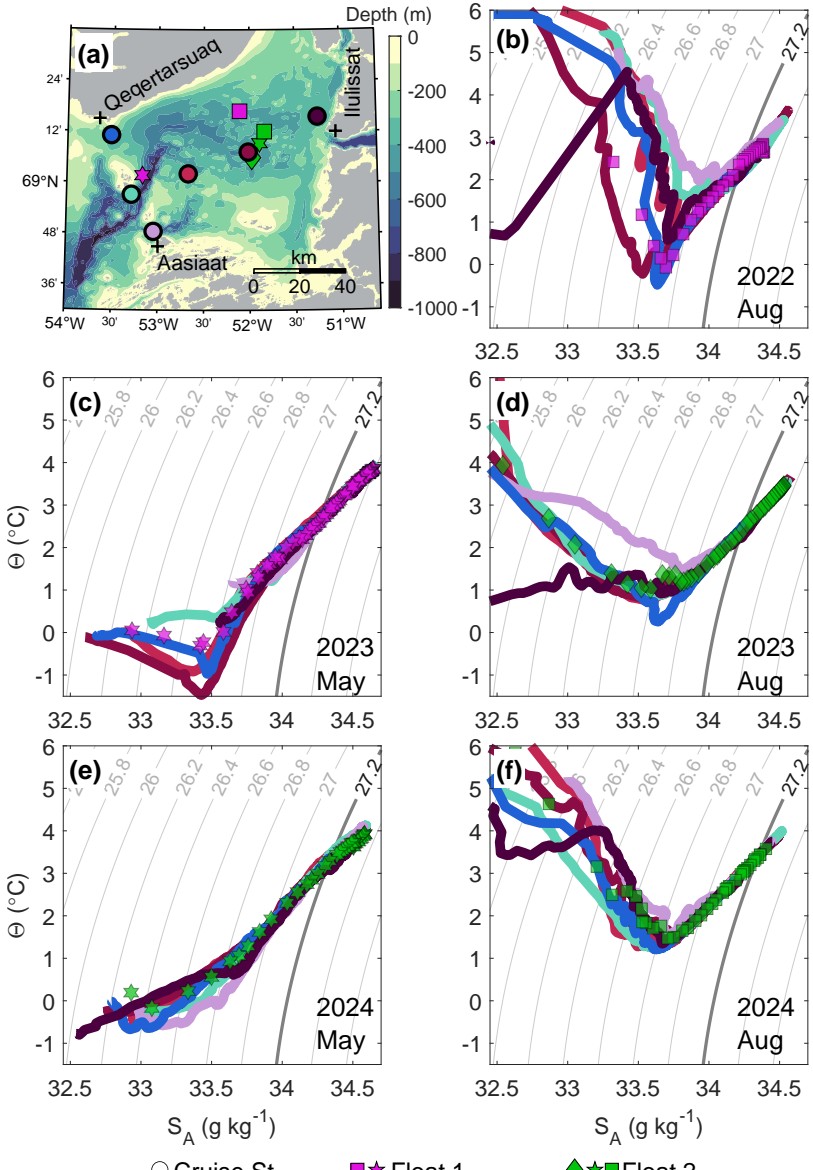

**Figure 4.** Spatial hydrographic variability across Disko Bay from GEM spatial surveys (2022–2024) and float observations. (a) Map of station locations (circle markers). Float 1 profiles are shown as magenta markers (square, pentagram) and Float 2 as green markers (diamond, pentagram, square). (b–f) Temperature–Salinity ($\Theta$–$S_A$) diagrams for each cruise. Left column: May (2023, 2024). Right column: August (2022, 2023, 2024). Coloured lines and float markers correspond to stations and colours shown in (a). Grey contours show isopycnals, with the thick grey contour delineating West Greenland Irminger Water (WGIW) and Polar Water (PW).

creases mixed-layer salinity through brine rejection (Fig. 5c). The maximum MLD occurs in winter, reaching 61 m in February 2023 and 48 m in January 2024. Continued sea-ice formation increases mixed-layer salinity to a peak of 30–33.4 g kg$^{-1}$ in

March (Fig. 5c), although enhanced stratification and elevated salinity below the mixed layer limit further deepening (Fig. 2e–f).

As sea ice starts to melt in late April, the mixed layer warms and freshens. This timing is consistent for both 2023 and 2024 and agrees well with satellite-derived SIC, which drops below 10% at that time (Fig. 5b–c). Sea-ice melt establishes a shallow mixed-layer (MLD < 20 m), which warms rapidly due to solar insolation (Fig. 5a–b). Through summer, increasing freshwater input lowers mixed-layer salinity by ∼2 g kg$^{-1}$, resulting in either a shallow or absent mixed layer (empty markers in Fig. 5a). By August–September, mixed-layer salinity reaches its minimum (31–31.4 g kg$^{-1}$), while temperature peaks at

8–10°C. (Fig. 5b–c).

    In autumn, the mixed layer cools and deepens, accompanied by a gradual increase in salinity, returning the system to winter conditions and completing the annual cycle.

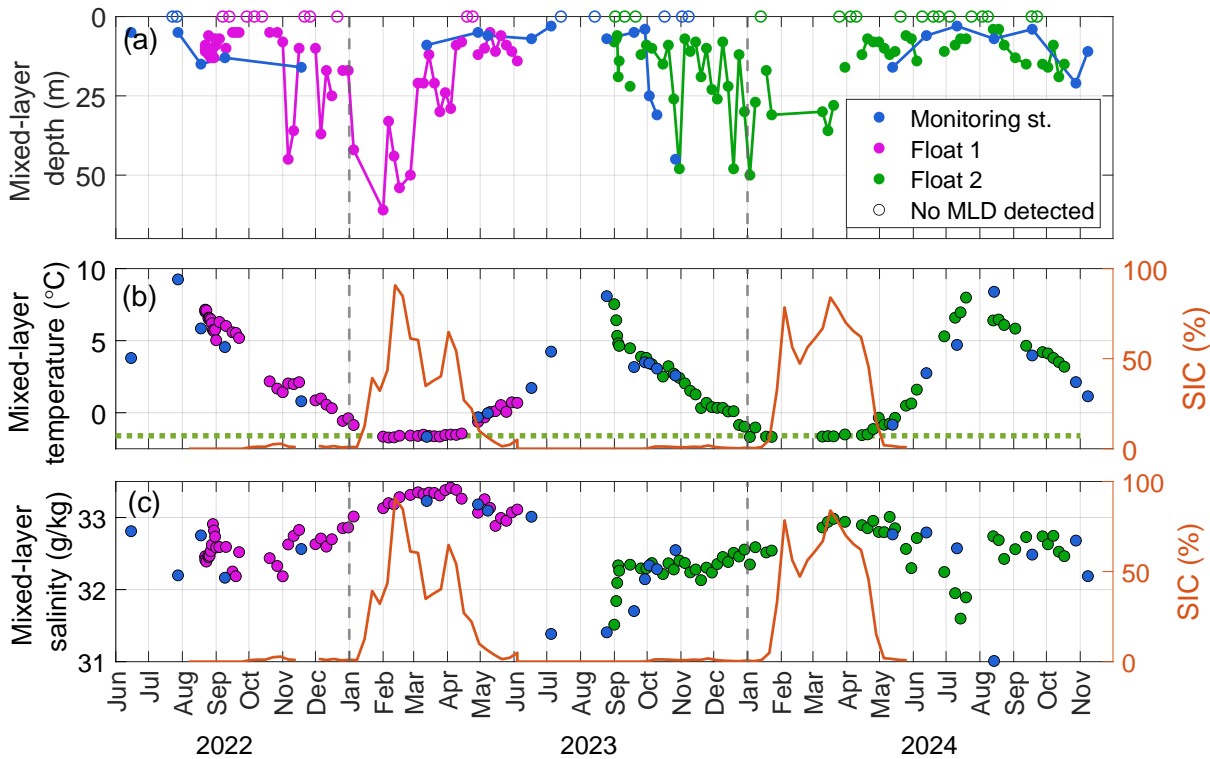

**Figure 5.** Time series of mixed-layer properties in Disko Bay from 2022 to 2024. (a) Mixed-layer depth (MLD), (b) mixed-layer temperature, and (c) mixed-layer salinity. Sea-ice concentration (SIC) within Disko Bay is shown in orange (right axis in (b–c)). Data are from the Monitoring Station (blue), Float 1 (magenta), and Float 2 (green). Empty circle markers in (a) indicate times when the mixed layer was not detectable.

## 4.4 Seasonality of Polar Water

At the onset of summer, near-surface salinity reduces in conjunction with mixed-layer freshening (Fig. 2c–d), establishing a
stratified layer with a core that is bound approximately by the $\sigma_0 = 26.5$ kg m$^{-3}$ isopycnal (Fig. 2e–f). As freshwater input and
surface warming intensify, the stratified layer thickens and extends to greater depths, reaching 50 m by late summer, and rapidly
extending to $> 100$ m during autumn (Fig. 2e–f). Although both the Floats and the Monitoring Station capture this seasonal
evolution, we assess PW seasonality primarily from the three-year Monitoring Station observations to avoid conflating the
substantial spatial variability in PW that Floats would capture (Section 4.2).

The mean salinity of the upper 120 m, encompassing the depth range of the stratified fresh layer ($\sigma_0 < 26.5$ kg m$^{-3}$),
shows a recurring seasonal reduction beginning in June (2023) or July (2022, 2024) and continuing until Monitoring Station
observations end in November (Fig. 6a). The magnitude of the summer–autumn freshening varies interannually, reaching
$\sim 0.5$ g kg$^{-1}$ in 2022, $\sim 1$ g kg$^{-1}$ in 2023, and $\sim 0.6$ g kg$^{-1}$ in 2024. Within the $\sigma_0 < 26.5$ kg m$^{-3}$ layer, temperature rises
through summer, peaks in August–September, and subsequently cools in autumn while salinity continues to decline (Fig. 6a–b).

Beneath this layer, denser PW ($\sigma_0 \approx 26.5$–$27.1$ kg m$^{-3}$) maintains the summer temperature minimum characteristic of PW
(Fig. 3d). In contrast to the lighter PW above (Fig. 6b), along-isopycnal temperatures within this denser PW layer continue to
increase steadily through autumn — rising by 1.6°C in 2022, 1.1°C in 2023, and 0.4°C in 2024 between August and November
(Fig. 6c).

In $\Theta$–$S_A$ space, this steady autumn warming of the denser PW ($\sigma_0 \approx 26.5$–$27.1$ kg m$^{-3}$) appears as a gradual erosion of its
sharp summer temperature minimum, with $\Theta$–$S_A$ properties shifting toward warmer values during autumn (Fig. 7a–c). Given
the strong spatial variability in August PW properties (Section 4.2; Fig. 4b, d, f), the warming at the Monitoring Station likely
reflects advection from upstream. In August, the station near Aasiaat, in the bay's southwestern corner, consistently had the
warmest PW, whereas the Monitoring Station had the coldest PW temperature minimum. When the August $\Theta$–$S_A$ properties
from Aasiaat are overlaid onto the Monitoring Station data, the latter's October–November properties closely resemble those
observed upstream two to three months earlier (Fig. 7a–c), consistent with cyclonic circulation and advection of water masses.
The $\sim 170$–$200$ km distance between Aasiaat and Monitoring Station implies mean velocities of $\sim 3.3$–$3.9$ cm s$^{-1}$ and $\sim 2.2$–
$2.6$ cm s$^{-1}$ for a two- and three-month lag, respectively.

The magnitude of autumn warming at the Monitoring Station along a given isopycnal matches the temperature differ-
ence between the Monitoring Station and the Aasiaat station along that same isopycnal in August. For example, along $\sigma_0 =$
$26.8$ kg m$^{-3}$, warming from August to November was 1.65°C in 2022, 1.1°C in 2023, and 0.48°C in 2024 along $\sigma_0 =$
$26.8$ kg m$^{-3}$ (Fig. 6c). These values are comparable to the corresponding along-isopycnal temperature offsets between the
Monitoring Station and Aasiaat station in August: 1.89°C in 2022, 1.1°C in 2023, and 0.56°C in 2024 (Fig. 7).

A similar relationship was found only once between the Monitoring Station and the station north of Ilulissat Icefjord. In
2022, denser PW properties at the Monitoring Station in September resembled those north of Ilulissat Icefjord in August. The
1.2°C warming along $\sigma_0 = 26.8$ kg m$^{-3}$ between August and September at the Monitoring Station (Fig. 6c) was comparable

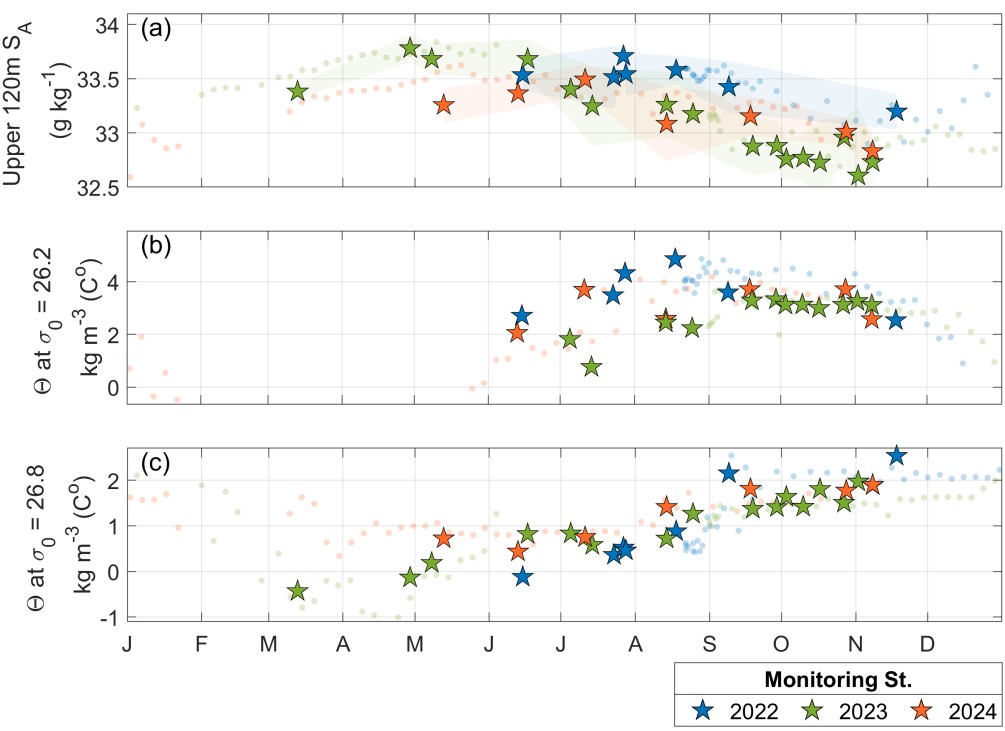

**Figure 6.** Seasonal cycle of (a) mean salinity in the upper 120 m (one standard deviation shaded in the background), (b) along-isopycnal temperature at $\sigma_0 = 26.2$ kg m$^{-3}$, and (c) along-isopycnal temperature at $\sigma_0 = 26.8$ kg m$^{-3}$. Monitoring Station observations for 2022, 2023, and 2024 are shown with blue, green, and orange star markers, respectively. Faint background markers show Float 1 and 2 observations, included here for reference.

to the 1.04°C temperature difference between the two locations in August 2022 survey (Fig. 7a). The distance of ∼90 km and a 1-month lag yield mean velocities of ∼3.5 cm s$^{-1}$.

### 4.5 Seasonality of West Greenland Irminger Water

At depths larger than 300 m, WGIW is isolated within Disko Bay by shallow bathymetric barriers (245 m at the entrance to
Vaigat Strait, 245 m at Ilulissat Icefjord, 300 m at EDS; Fig.1b). Regarded as basin water, WGIW can be renewed if equally dense or denser waters pass over the topographic constraints (Gade and Edwards, 1980), with the deepest and most relevant being the EDS (Andersen, 1981a; Gladish et al., 2015a). The onset of a WGIW renewal is characterised by an increase in density below 300 m.

In early November 2022, basin density at 400 m increased from a September-October mean of $\sigma_0(400m)=$ 27.31 kg m$^{-3}$ to
27.36 kg m$^{-3}$ by early December (grey shading in November–December in Fig. 8e). Basin temperatures rose simultaneously by ∼ 0.3°C (Fig. 8f). Around two weeks after the onset of density increase, the upper WGIW boundary ($\sigma_0 = 27.2$ kg m$^{-3}$) shoaled rapidly by >100 m within a span of two weeks (Fig. 8d), reflecting the uplift of lighter WGIW that previously resided

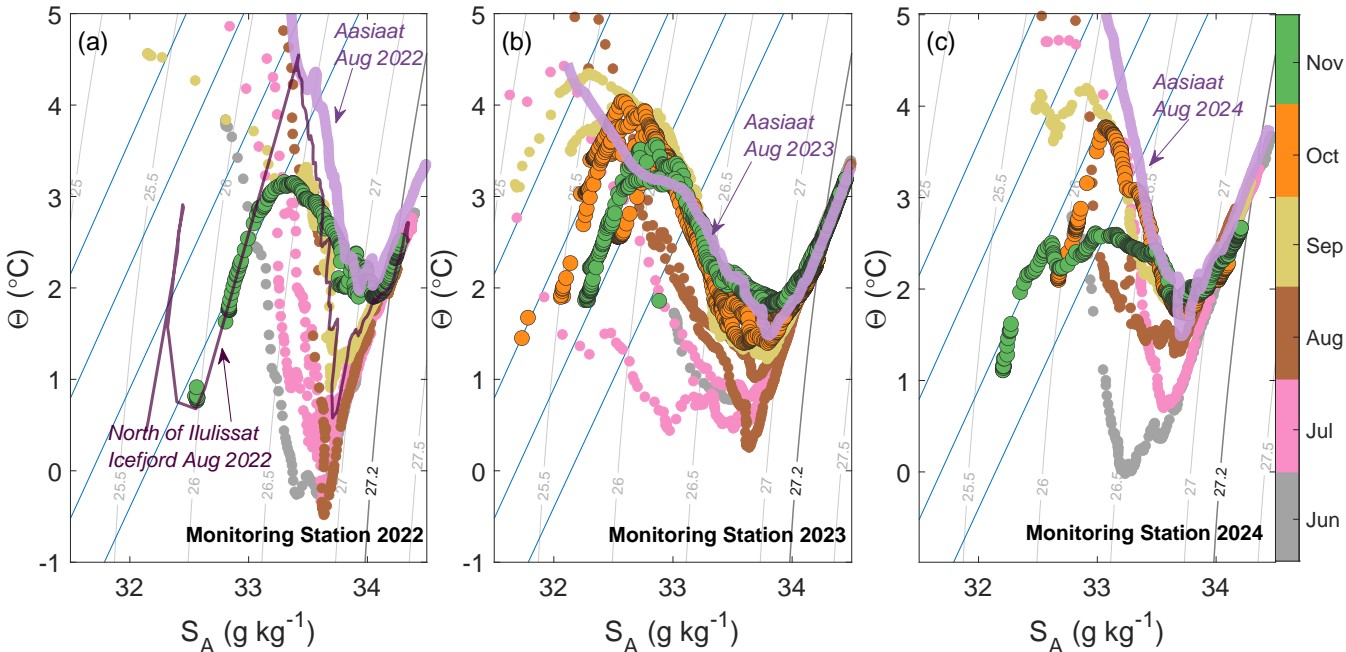

**Figure 7.** Temperature - Salinity ($\Theta$-$S_A$) diagrams for the Monitoring Station for (a) 2022, (b) 2023, and (c) 2024, covering June–November. Circle markers are observations coloured by month; October-November are shown with a black outline. The thick lilac line in panels (a–c) shows $\Theta$-$S_A$ properties at the station near Aasiaat during August of the same year (previously shown in Fig.4b, d, f); the thin purple line in (a) is the same, but for the station north of Ilulissat Icefjord. Isopycnals are shown in grey, with the thick contour delineating West Greenland Irminger Water (WGIW) and Polar Water (PW). Submarine meltwater ($\Theta$ = -90°C, $S_A$ = 0 g kg$^{-1}$) mixing lines are shown in blue

in the basin. This renewal coincided with a seasonal shift in prevailing wind direction, whereby the along-coast (north–south) winds switched from being predominantly southerly in summer to northerly in autumn to spring (from positive wind stress values to negative in Fig. 8b). In the absence of strong sea-ice cover, northerly winds (negative wind stress) drive upwelling over the Egedesminde Dyb and its sill (Egedesminde Dyb Sill, EDS). Using hourly data, we define "strong upwelling" as $W_E \geq 0.45$ m day$^{-1}$ (upper quartile of the hourly data distribution), typically derived under $\tau_y \leq -0.06\,\mathrm{N\,m}^{-2}$. November 2022 exceeded these thresholds with $\overline{\tau}_y = -0.1$ N m$^{-2}$ and $\overline{W}_E \geq 0.7$ m day$^{-1}$ (Fig. 8b, c); $W_E$ frequently exceeded $1\,\mathrm{m\,day}^{-1}$ (upper quartile), with episodic peaks approaching $3\,\mathrm{m\,day}^{-1}$. The strength and persistence of this forcing appear to have lifted the dense waters to the west over EDS, initiating the observed basin renewal in November-December 2022 (Fig. 8d–f).

Through winter 2022–2023, basin density continued to rise gradually from $\sigma_0(400m)$= 27.36 kg m$^{-3}$ to 27.39 kg m$^{-3}$ between February and late April (Fig. 8e), accompanied by a further $\sim 0.5°C$ increase in basin temperature which reached the annual peak at the end of April 2023 (Fig. 8f). This dense renewal lifted the overlying WGIW, which continued to rise until early June, when the upper WGIW boundary ($\sigma_0 = 27.2$ kg m$^{-3}$) rose to a minimum depth of 120 m (Fig. 8d). Northerly winds ($\overline{\tau}_y = -0.04$ to $-0.07\,\mathrm{N\,m}^{-2}$, upper quartile $\leq -0.12\,\mathrm{N\,m}^{-2}$), Ekman divergence, and positive vertical velocities

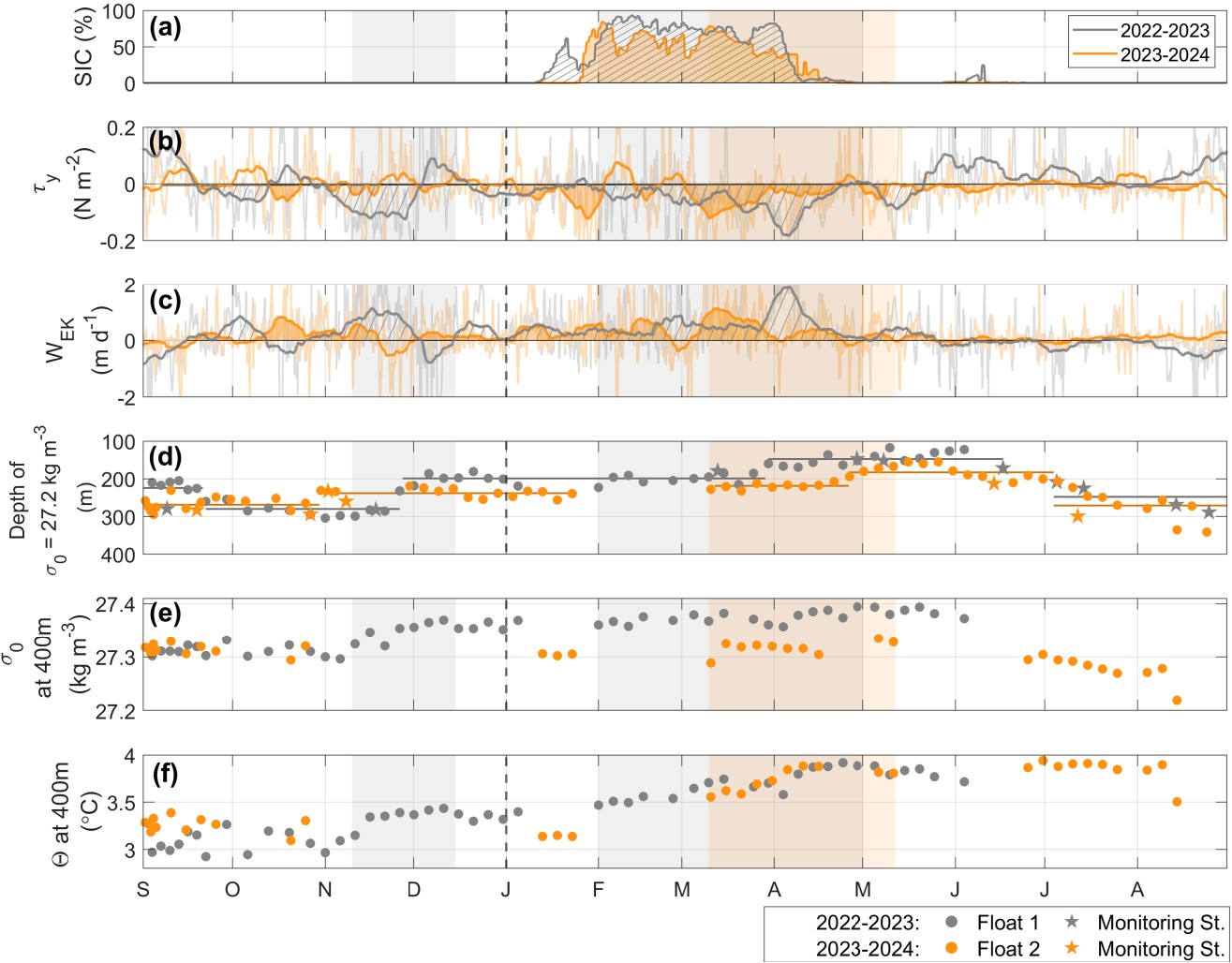

**Figure 8.** Seasonality of atmospheric forcing and WGIW properties in Disko Bay over two annual cycles (2022–2023 in grey, 2023–2024 in orange, circle markers for the Floats, star markers for the Monitoring Station). Mean sea-ice cover (a), along-shore wind stress ($\tau_y$) (b), and vertical velocity $W_E$ (Ekman pumping) over the EDS area (Fig. 1a) (c). In panels (b–c), hourly data are overlaid with a 10-day running mean (thick lines); shaded and hatched areas highlight the negative and positive ranges in (b) and (c), respectively. Depth of the $\sigma_0$= 27.2 kg m$^{-3}$ — upper WGIW boundary (d), with horizontal lines indicating the mean depth before abrupt changes, identified using MATLAB findchangepts() function (Lavielle, 2005; Killick et al., 2012). Density (e) and temperature (f) at 400 m depth from Float 1 and Float 2. Basin renewal periods are shaded in all panels. The dashed vertical line in all panels marks a new year.

($\overline{W}_E = 0.47 - 0.82\,\mathrm{m\,day^{-1}}$, upper quartile $\geq 1.5\,\mathrm{m\,day^{-1}}$) persisted through February–April until the prevailing winds reversed in mid-May, ending the upwelling-favourable conditions (Fig. 8b–c).

In autumn 2023, the upper WGIW boundary ($\sigma_0 = 27.2$ kg m$^{-3}$) shoaled by ∼50 m in early November (Fig. 8d). There

were no Float observations deep enough in the basin to document subsequent changes until early winter. However, as neither density nor temperature increased from October 2023 until January 2024, basin renewal likely did not occur. The average wind stress was near zero from September to December, with brief episodes of negative $\tau_y$ in October and November (Fig. 8b). Calculated upwelling velocities were $\overline{W}_E = 0.3$ m day$^{-1}$ in October, with short-lived episodes of $W_E > 0.6$ m day$^{-1}$ (upper quartile), but overall, the forcing was weaker and less persistent than in November 2022 (Fig. 8c).

There are signs of a renewal in March–April 2024, when $\sigma_0(400m)$ increased by 0.01 kg m$^{-3}$ and the temperature rose by ∼ $0.3°C$ within one month (orange shadings in Fig. 8e, f). Lack of observations between 23 January and 11 March hinders the ability to determine when this renewal began; however, density and temperature in the basin were already higher by 0.02 kg m$^{-3}$ and ∼ $0.4°C$ by 11 March, suggesting it was already underway. Winds during January–April were upwelling-favourable ($\overline{\tau}_y$ = -0.01 to 0.05 N m$^{-2}$), with mean vertical velocities of $0.1-0.6$ m day$^{-1}$. The strongest forcing (upper quartile)

occurred in March with $\tau_y \leq -0.11$ N m$^{-2}$ and $W_E \geq 1.1$ m day$^{-1}$. By late May – early June 2024, the WGIW boundary rose to a minimum depth of ∼150 m (Fig. 8d).

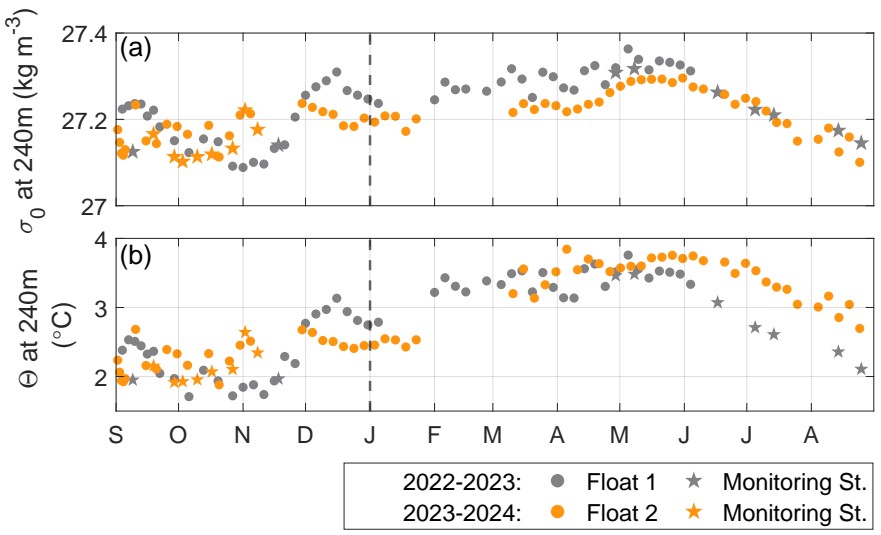

**Figure 9.** Seasonal cycle of (a) density and (b) temperature at 240 m depth, corresponding to the Ilulissat Icefjord Sill depth. Grey markers are observations from 2022–2023, and orange from 2023–2024. Star markers are for the Monitoring Station observations, and circles for the Floats 1 and 2.

In both years, the upper WGIW boundary was shallowest in May–June, shoaling to 120 m in 2023 and 150 m in 2024 (Fig. 8d). This uplift brought WGIW waters in Disko Bay above 240 m depth, corresponding to the Ilulissat Icefjord sill depth. At 240 m depth, density showed a pronounced seasonal cycle, peaking at $\sigma_0$= 27.3-27.35 kg m$^{-3}$ in May and declining to a

minimum of $\sigma_0$= 27.1 kg m$^{-3}$ in August–November (Fig. 9a). Temperature at 240 m varied in parallel, with an amplitude of

$\sim 2°C$ and maxima above $3.5°C$ in May–June (Fig. 9b). A renewal event in November–December 2022 resulted in $\sim 1°C$ warming at 240 m (Fig. 9b).

## 5  Discussion

Our results documented two full annual cycles (2022–2024) of hydrographic properties within Disko Bay. Beyond the expected seasonality in the surface mixed layer (Fig. 5), we observed notable seasonal and spatial variability within PW and a recurring inflow of dense WGIW that replenishes the deep basin and elevates the WGIW–PW interface. Our results reflect the renewed "warm" state of Disko Bay that has persisted since 2020 (Picton et al., 2025), following the short-lived anomalously cool period of 2015–2019 (Khazendar et al., 2019; Picton et al., 2025).

The seasonal cycle of PW can be divided into three phases: (1) in winter–spring, cooling, sea-ice formation, and brine rejection increase the density and depth of the mixed layer; (2) with the onset of the melt season, a fresh stratified layer develops in the upper 50 m, bounded by the $\sigma_0 = 26.5$ kg m$^{-3}$ isopycnal (Fig. 2e); and (3) in late summer–autumn, stratified layer ($\sigma_0 < 26.5$ kg m$^{-3}$) extends over 120 m while freshening and cooling (Fig. 6a–b), while the denser PW beneath ($\sigma_0 \approx$ 26.5–27.1 kg m$^{-3}$) warms steadily along isopycnals (Fig. 6c; Fig. 7). In the following sections, we examine the mechanisms driving this autumn evolution of PW before turning to the renewal of WGIW.

### 5.1  Autumn evolution of Polar Water

#### 5.1.1  Remote and local drivers of autumn freshening

Both remote and local processes can contribute to the autumn freshening observed in the upper 120 m (Fig. 6a) and the associated vertical expansion of the stratified layer (Fig. 2e). The primary remote source is advection of signals from the West Greenland Current (WGC). Observations from the Davis Strait show a clear annual cycle in salinity, with peak salinities along the west Greenland shelf between April–June (33.66 g kg$^{-1}$) and a decrease towards the annual minimum in August–October (32.75 g kg$^{-1}$) (Curry et al., 2014; Carroll et al., 2018). Similar seasonality occurs along the shelf-slope at $\sim 150$ m (Gladish et al., 2015b). Given that Davis Strait lies $\sim 400$ km south of Disko Bay and the subsurface WGC velocity is $\sim 0.07$ m s$^{-1}$ in autumn (Curry et al., 2014), these signals could reach Disko Bay with a delay of about two months, consistent with the timing of autumn freshening at the Monitoring Station (Fig. 2d,f; Fig. 6a). However, minimum density and salinity at $\sim 150$ m in Davis Strait tend to be higher ($\sigma_0 > 27$ kg m$^{-3}$ and $S_A$>33.8 g kg$^{-1}$)(Gladish et al., 2015b) than those observed at the same depth in Disko Bay during autumn (Fig. 2c–d), suggesting that local freshwater inputs within Disko Bay likely contribute as well.

The significant local source is Sermeq Kujalleq, which delivers large summertime freshwater fluxes through subglacial discharge (SGD, 900 m$^3$s$^{-1}$ (Mernild et al., 2015; Enderlin et al., 2016) and submarine meltwater (SMW, 70-400 m$^3$s$^{-1}$ (Enderlin et al., 2016; Kajanto et al., 2023). A mix of these melt products, together with entrainment of PW and WGIW, forms GMW, whose vertical reach is strongly controlled by SGD seasonality (Cowton et al., 2023; Kajanto et al., 2023; Hager

et al., 2024). Icebergs within Ilulissat Icefjord are a dominant freshwater contributor (700–1000 $m^3s^{-1}$ in winter, up to 1200–1800 $m^3s^{-1}$ in August) (Enderlin et al., 2016; Kajanto et al., 2023), altering water mass properties within the fjord, modifying the neutral buoyancy depth of GMW, and cooling the fjord basin through reflux of outflowing water (Kajanto et al., 2023;

Hager et al., 2024). The GMW exported from the fjord over the sill flows north in Disko Bay as a buoyant stratified current, with a cold, fresh signature extending to at least 100 m depth near the shore (Beaird et al., 2017). Offshore, this layer shoals to $\sim 35$ m, but retains its signature up to 10 km offshore (Beaird et al., 2017). Our spatial analysis detected similar anomalies each August north of Ilulissat Icefjord (Fig. 2b, d, f). Given that the peak melt season in the fjord occurs in July–August (Wood et al., 2025; Picton et al., 2025; Mernild et al., 2015; Enderlin et al., 2016; Kajanto et al., 2023), the export of thick meltwater-laden

GMW could contribute to the freshening observed downstream at the Monitoring Station during autumn. Continued iceberg melt after the melt season (Moon et al., 2018; Kajanto et al., 2023) may also explain why autumn observations in the $\Theta$–$S_A$ diagram aligned with the SMW mixing line (Fig. 3a–c; Fig. 7a–c). While such alignment could reflect inputs of iceberg melt, it might also result coincidentally from autumn cooling, and disentangling these processes would require additional tracers (Beaird et al., 2015, 2018; Lindeman et al., 2024).

Finally, the iceberg melt within Disko Bay itself provides an additional but smaller freshwater input. The annual average solid ice discharge at the terminus of Sermeq Kujalleq is $\sim 50$ Gt yr$^{-1}$, equivalent to $\sim 1500$ $m^3s^{-1}$ (Mankoff et al., 2020). Given that the annual average iceberg melt inside the fjord is $\sim 1200$ $m^3s^{-1}$ (Kajanto et al., 2023), up to 80% of discharged icebergs likely melt inside Ilulissat Icefjord. The remaining fraction can cross the sill and enter Disko Bay, where more than 1000 small icebergs (with an area of about 1800 $m^2$) can be observed simultaneously (Scheick et al., 2019). To estimate their potential

freshwater flux, we represent an average iceberg of this size as $\sim 130$ m in length and width. Icebergs in Ilulissat Icefjord are typically twice as wide as they are thick (Enderlin et al., 2016), giving an estimated thickness of $\sim 65$ m (freeboard+draft). Using the freeboard-to-draft ratio of 1:7 (Cenedese and Straneo, 2023), we estimate the draft of $\sim 55$ m. Given the average summertime temperature of 2°C in the upper 50 m, and assuming fully turbulent conditions around the iceberg, we estimate that the upper bound of summertime meltwater flux from 1000 of such icebergs would be around 65 $m^3s^{-1}$. While non-negligible,

this contribution is small compared to freshwater inputs within Ilulissat Icefjord and therefore unlikely to be the primary driver of the continued autumn freshening observed at the Monitoring Station.

### 5.1.2   Along-isopycnal warming at depth

Beneath the stratified fresh layer, a continued along-isopycnal warming was observed within denser PW ($\sigma_0 \approx 26.5$–$27.1$ kg m$^{-3}$; Fig. 6c; Fig. 7). By October–November, hydrographic properties at the Monitoring Station closely matched those measured

near Aasiaat two to three months earlier, suggesting an advective signal that may enter Disko Bay from the southwest and be advected cyclonically around the bay.

A similar late-autumn warming below 150 m was previously documented by Hansen et al. (2012), who attributed it to entrainment of warm surface waters. In our observations, however, the overlying PW layer cools and continues to freshen during this period, while the warming at depth is accompanied by a slight increase in salinity (Fig. 7b,d,f). These features are

more consistent with the advection of warmer, saltier water masses than with vertical mixing from the surface.

Without hydrographic profiles outside Disko Bay, we cannot definitively establish the sources of this warming before it appears near Aasiaat. However, at depths comparable to those of the $\sigma_0 \approx 26.5$–$27.1$ kg m$^{-3}$ PW layer in autumn (Fig. 2), long-term moorings at Davis Strait record a pronounced seasonal temperature cycle along the west Greenland shelf and slope. Instruments at 151 m and 252 m show temperatures increasing from summer minima to peak values in December-February, while water masses remain least dense during autumn (Gladish et al., 2015b; Carroll et al., 2018). Although mooring records are not yet available for our study period, the warming observed propagating toward the Monitoring Station in autumn may represent the advection of this recurring Davis Strait signal.

Glacial processes may also contribute to the observed warming. Buoyant plumes of SGD and SMW drive turbulent upwelling that entrains the warm and saline Atlantic-origin waters, producing GMW that is commonly warmer and more saline than unmodified Polar Water of the same density found at a distance away from the glacier (Straneo et al., 2012; Beaird et al., 2017, 2018; Mortensen et al., 2020; Muilwijk et al., 2022; Cowton et al., 2023). For instance, GMW exported from Ilulissat Icefjord has been estimated to contain about 40% of WGIW (Beaird et al., 2017), making it distinctly warmer and saltier than ambient PW. GMW export from Ilulissat Icefjord may therefore enhance autumn anomalies in PW. However, our spatial analysis showed that the highest along-isopycnal temperatures in August occurred further upstream, near Aasiaat (and, in 2023, within the deep trough entering Disko Bay), rather than near Ilulissat Icefjord. We therefore interpret the autumn along-isopycnal warming primarily as the seasonal signal of the WGC, while GMW exported from Ilulissat Icefjord may provide a secondary, but smaller, contribution.

## 5.2 West Greenland Irminger Water renewal

Over two annual cycles (2022–2023, 2023–2024), renewal of WGIW in Disko Bay occurred primarily in spring, when the densest WGIW filled the basin, resulting in annual maxima in density, temperature, salinity, and vertical extent of WGIW (Fig. 8d–f). The springtime renewal began in February–March and lasted until May–June. This provides direct observational evidence supporting the hypothesis of Gladish et al. (2015a, b), who proposed that exchange over the EDS likely occurs in spring. This timing also matches observations further north in Uummannaq fjord system (Carroll et al., 2018), and south in Godthåbsfjord (Mortensen et al., 2011, 2014). At 240 m in Disko Bay (Ilulissat Icefjord sill depth equivalent), WGIW density peaks at $\sigma_0 = 27.3$–$27.35$ kg m$^{-3}$ in May (Fig. 9a), matching the density range observed within Ilulissat Icefjord basin (Gladish et al., 2015b, a; Beaird et al., 2017). This correspondence suggests that spring renewal in Disko Bay delivers the densest waters entering the fjord basin.

The repeated springtime renewal in Disko Bay is consistent with regional-scale hydrographic variability. At Davis Strait, isopycnals tilt up toward the Greenland shelf in spring, with ∼140 m vertical displacement of WGIW at the shelfbreak co-incident with densification along the shelf (Curry et al., 2011, 2014; Gladish et al., 2015b; Carroll et al., 2018). Increasingly dense waters thus become available over the EDS, eventually exceeding the density of resident basin waters in Disko Bay and driving the springtime renewal (Fig.10b). Persistent upwelling-favourable winds locally through spring likely enhance the renewal process (Fig.8b–c).

In addition to the repeated spring renewal, we observed a distinct renewal in November–December 2022, marked by rapid increases in basin density and temperature and a >100 m shoaling of the WGIW boundary (Fig. 8d–f). This autumn/winter renewal was unique to the 2022–2023 annual cycle and coincided with particularly strong and persistent upwelling-favourable winds. These conditions likely induced upwelling over the EDS, lifting the dense waters over the sill and into Disko Bay (Fig.10a). Estimated vertical velocities frequently exceeded $1\,\mathrm{m\,day^{-1}}$, with episodic peaks approaching $3\,\mathrm{m\,day^{-1}}$, implying uplift on the order of $\sim$20 m during this period. The actual uplift may have been greater, as our estimates only quantify the effect of wind stress curl, and not coastal upwelling, which could also contribute, given the proximity of the coastline east of EDS.

While we cannot determine the precise magnitude of upwelling or the properties of upwelled waters over the EDS without hydrographic observations on the shelf, the conditions observed are consistent with other studies linking upwelling-favourable winds to WGIW intrusions along the west Greenland shelf. For example, upwelling-favourable winds were linked to the observed increased WGIW presence in the Uummannaq trough (300 km north of Disko Bay) during December–January (Carroll et al., 2018), and to modelled areas of frequent upwelling along the coast both north and south of Disko Bay (Ribergaard et al., 2004; Söderkvist et al., 2006). Near Cape Farewell, wind-driven upwelling events have been shown to draw Atlantic-origin waters from $\sim$ 250 m depth onto the shelf ($\sim$ 150 m), raising both temperature and salinity (Pacini and Pickart, 2023). In southeast Greenland, at Sermilik Fjord, along-shelf winds were also shown to play an important role in uplift and onshore intrusion of dense Atlantic Waters (Snow et al., 2023; Sanchez et al., 2024).

Our results highlight that a pronounced autumn/winter WGIW renewal, such as in 2022, can shoal the WGIW boundary enough to increase the temperatures at Ilulissat Icefjord sill depth by $\sim 1^\circ C$ ( Fig. 9b). While fjord renewal is primarily driven by SGD during the melt season (Gladish et al., 2015b, a; Carroll et al., 2017), our observations show that episodic autumn/winter uplift of dense waters in Disko Bay can make waters denser than those in the fjord basin available above the sill earlier in the year than previously expected (Gladish et al., 2015a, b) and during periods of limited SGD forcing (Mernild et al., 2015; Enderlin et al., 2016). Observations in Disko Bay reported by Picton et al. (2025) indicate that late autumn–early winter renewal may have taken place in 2021 as well, as temperatures at 240 m increased rapidly by $\sim 0.5^\circ C$.

We therefore extend the schematic of Gladish et al. (2015a) to include a wind-driven autumn/winter renewal pathway across the EDS into Disko Bay, while also acknowledging the possibility that such events may enable dense inflow into Ilulissat Icefjord basin (Fig.10a), although direct evidence for this process is not yet available.

## 6 Conclusions

We examined two full annual cycles (June 2022 – November 2024) of Disko Bay hydrography using fixed-point observations from a monitoring station, two profiling floats, and near-synoptic surveys to resolve the seasonal evolution and spatial structure of PW and WGIW.

Sea-ice melt initiates a shallow mixed-layer that warms up to $\sim$ 8–10°C while freshening to $\sim$ 31–31.4 g kg$^{-1}$ by late summer. Spatially, the surface layer varies across the bay, with the coldest and freshest waters found near Ilulissat Icefjord.

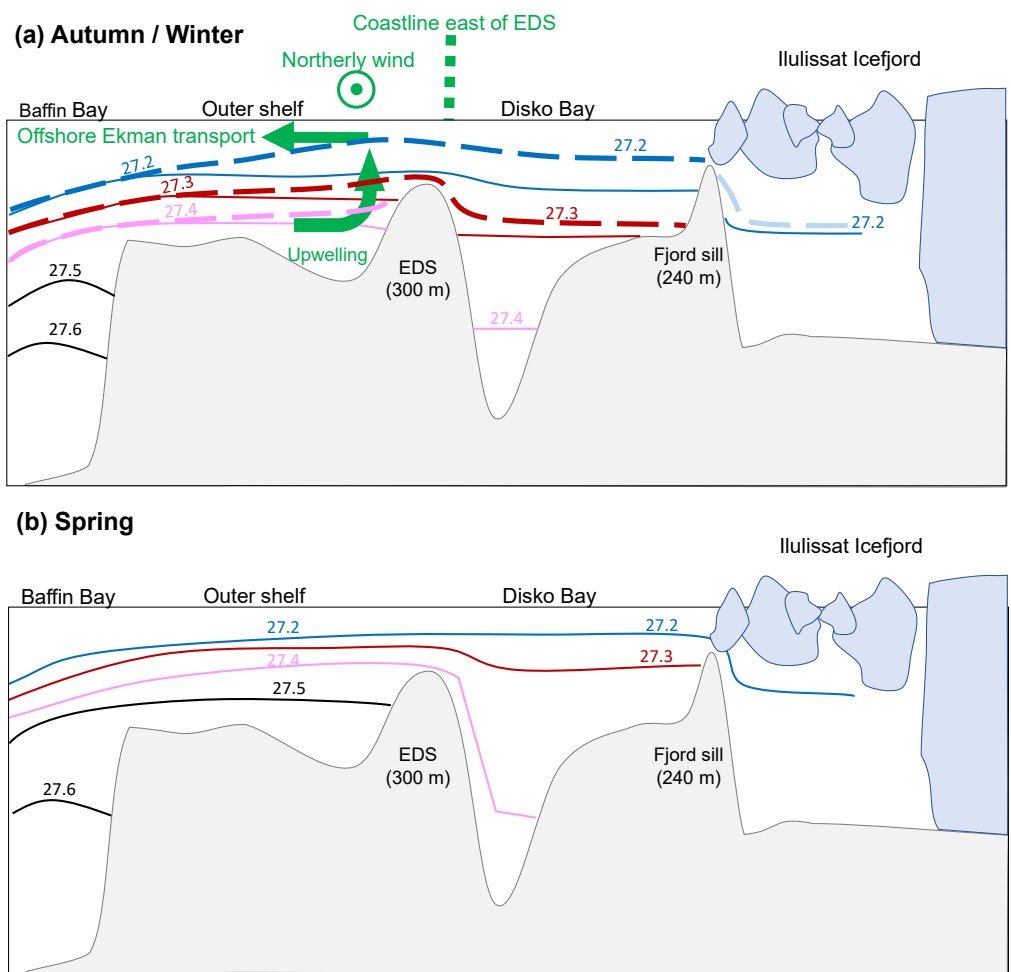

**Figure 10.** Schematic of West Greenland Irminger Water (WGIW) exchange between the west Greenland Shelf, Disko Bay, and Ilulissat Icefjord, adapted from Gladish et al. (2015a). Solid isopycnals and bathymetry in (a–b) are reproduced from the original schematic. Panel (a) has been updated to illustrate a potential autumn/winter wind-driven renewal mechanism: Northerly along-shore winds (negative wind stress, ⊙) drive offshore Ekman transport and upwelling (green arrows), lifting dense isopycnals over the Egedesminde Dyb Sill (EDS) and allowing renewal of Disko Bay (dashed lines), and possibly Ilulissat Icefjord (fainter dashed line). The coastline east of EDS is indicated as it likely enhances wind-driven upwelling. Panel (b) shows spring renewal process, when seasonal densification of WGIW on the west Greenland Shelf raises isopycnals above sill depth, enabling Disko Bay and Ilulissat Icefjord renewal.

Below, freshwater input establishes a stratified layer bounded by $\sigma_0 \approx 26.5$ kg m$^{-3}$, which progressively thickens and extends downward from the upper $\sim 50$ m in summer to depths exceeding 100 m in autumn. This layer cools and continues to freshen during autumn. Along isopycnals within denser PW ($\sigma_0 \approx 26.5$–27.1 kg m$^{-3}$), temperatures steadily increase through autumn. Near-synoptic surveys show that the warmest August PW of such densities occurs upstream at the southwestern end of the bay, and that Monitoring Station properties in October–November match those found upstream with a 2–3 month delay, consistent

with cyclonic advection around the bay. PW exhibits strong spatial variability, supporting the use of the Monitoring Station as the site for studying PW seasonality. In contrast, WGIW properties fall along a narrow $\Theta$–$S_A$ line, and spatial variance along depth levels is small below 300 m, supporting the use of float data to characterise WGIW seasonality.

At depths > 300 m, WGIW is isolated by bathymetry and is renewed when denser waters cross Egedesminde Dyb Sill (EDS). Our observations reveal that WGIW is renewed annually. Renewal occurs in the spring, with an additional episode of renewal also observed in late autumn 2022. During spring, the densest WGIW fills the Disko Bay basin, peaking in temperature, salinity, and vertical extent by late spring/early summer. The seasonal springtime renewal, previously suggested by Gladish et al. (2015a), is confirmed by our observations. Its timing is consistent with regional isopycnal uplift and densification along

the west Greenland shelf. The distinct autumn/winter renewal in November–December 2022 coincided with unusually strong and persistent northerly winds over the EDS, which likely lifted denser waters over the topographic barrier, enabling the renewal. This suggests an additional mechanism and timing of intermittent WGIW renewal in Disko Bay.

    Overall, Disko Bay hydrography reflects the superposition of seasonal signals within WGC, spatially heterogeneous local freshwater inputs, local air- and ice-ocean interactions, and episodic wind-driven exchanges across EDS. The improved under-

460 standing of seasonality and spatial context provides a baseline for interpreting and predicting variability relevant to ice-ocean coupling and ecosystem dynamics within Disko Bay.

*Data availability.*   The merged MODIS-AMSR2 sea-ice concentration data are available at: https://data.seaice.uni-bremen.de/modis_ amsr2. Greenland Ecosystem Monitoring (GEM) data are available at: https://data.g-e-m.dk/datasets. Additional observations collected for this study at the Monitoring Station in 2023 will be available through the GEM database. ERA5 data are available at the Copernicus Climate

Change Service (C3S) Climate Data Store (CDS) (https://doi.org/10.24381/cds.adbb2d47, Copernicus Climate Change Service, Climate Data Store, 2023). Bathymetry data are available from the NASA National Snow and Ice Data Center Distributed Active Archive Center at: https://doi.org/10.5067/849. Oceans Melting Greenland Data for the profiling floats are available at: https://podaac.jpl.nasa.gov/dataset/OMG_L1_FLOAT_ Greenland Ocean Observations Apex-float data are available at: https://fleetmonitoring.euro-argo.eu/float/6990591.

*Author contributions.*   LL: Writing – review and editing, Writing – original draft, Visualisation, Methodology, Investigation, Formal analysis,

Conceptualisation. LH: Writing - review and editing, Writing – original draft, Methodology, Investigation, Conceptualisation, Supervision. ED: Writing - review and editing, Writing – original draft, Visualisation, Methodology, Investigation. PJH: Writing - review and editing, Resources, Data curation. JKW: Writing - review and editing, Resources, Data curation

*Competing interests.*   The authors declare that they have no conflict of interest.

*Acknowledgements.* We thank Dana Margareta King for processing the sea-ice data, Iliana Vasiliki Ntinou for assisting with the fieldwork,

and Torkel Gissel Nielsen for support with planning the fieldwork. We gratefully acknowledge the crew of RV *Porsild* and the staff at the Arctic Station (University of Copenhagen) in Qeqertarsuaq for their hard work and expertise in safely and effectively carrying out the fieldwork. We thank the NASA OMG Mission, as well as the GEM programme, for making observational data freely available. We acknowledge the collaborative efforts of scientists at the Greenland Institute of Natural Resources in collecting the NASA and NOAA "Greenland Ocean Observations" (GOO) float data that contributed to this study. This work was carried out in part at the Jet Propulsion

Laboratory, California Institute of Technology, under a contract with the National Aeronautics and Space Administration (80NM0018D0004). This work was supported financially by the Research Council of Norway through the project ClimateNarratives (no. 324520).

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
