# Peer review of "Drivers of seasonal hydrography in Disko Bay, Greenland"

_EGUsphere, 2025_

## Author Comment (AC1)

**Response to Reviewer#1**

We thank you for your constructive and thoughtful review of our manuscript. Your suggestions have been invaluable in improving the quality and clarity of our work. In particular, your feedback prompted several substantial revisions: (1) replacing the 2018 dataset with newly available spatial surveys from summers 2022-2024, (2) conducting a thorough spatial analysis based on these new surveys, and (3) incorporating the newly available Monitoring Station observations from 2024. These revisions were central to addressing your major comments and have considerably strengthened the manuscript.

Please find our detailed responses in blue below. In the few cases where it was not possible to follow your suggestions directly, we provide a detailed explanation in our responses.

Best regards,

Linda Latuta on behalf of all co-authors.

**Reviewer#1 Summary**

This paper presents observational hydrographic data from Disko Bay, Greenland, which sits near a large, fast tidewater glacier draining the Greenland Ice Sheet (Sermeq Kujalleq). They combine existing data sources from a nearby monitoring station (GEM site), two profiling floats, and re-analysis atmospheric & satellite data over two seasonal cycles (the years 2022-2024) along with weekly repeat stations taken over Fall 2023. In the end, they describe a seasonal cycle that is consistent over both years, showing both the modulation of the mixed layer and when the warmest and densest Atlantic-origin waters arrive at the mouth of Ilulissat Icefjord (which leads to Sermeq Kujalleq, SK hereafter). Overall, they present a detailed account of the hydrography in the bay's water mass layers, which should be beneficial to future studies looking to oceanic change as a driver for mass loss from Greenland. Although many pieces of the study are already known, I find the most novel results to be their discussion of the deep-water renewal and lifting of the Atlantic-origin waters above sill depth at the mouth of the icefjord. However, the paper seems overly complicated for the conclusions it reaches, and the figures could be clarified and improved given all the disparate sources of data. For example, explicitly stating what is novel here and what is validation of previous conceptual models of the Bay's exchange with the coast and the icefjord would be helpful. One way to do that would be to update or revise the schematic from Gladish et al. part 2 (his Fig. 15). But there are other ways too. I was also not convinced of the Fall along-isopycnal warming signal they attribute to outflowing glacially modified waters (GMW). Finally, a new paper (Picton et al. 2025) came out in JGR-Earth Surface recently that uses some of the same data (floats and GEM site), although their focus is slightly different.

More on these comments below, as well as line by line comments. I certainly believe these data and this write-up to be publication worthy, but their impact will be significantly improved through revisions.

**Major comments:**

1) During the discussion period, another paper came out that studies the same region with some of the same data sources (Picton et al. 2025). It would be good to include this reference in the

revision. For example, that paper shows temperature vs time at 240 m depth and seems to show a similar Fall warming and seasonality as described in this current study. I think discussion of this new publication will only strengthen this one. In terms of other references, I believe the authors could certainly expand their sources. I point to a few examples in the comments below.

This was a helpful suggestion, and we also believe the new Picton et al. (2025) provides valuable context for our study and strengthens both the Introduction and Discussion. In the revised manuscript, we have incorporated this reference and expanded our literature base to provide a more comprehensive context for our findings.

2) In terms of the along-isopycnal warming trend they document and discuss, I am not convinced it's entirely from the outflow from the icefjord. I think what they are saying is that the summer outflow signal is delayed and lagged as it transits Disko Bay, so the warming they see through the Fall is from that entrained WGIW. However, the melt season at the glacier (e.g., look at a subglacial discharge time series for SK) is relatively short and the warming seems to continue beyond the length of a typical melt season. Although they calculate advective time scales for other processes in the paper, they don't estimate the time scales here. Are there other mechanisms that could cause this warming trend through the Fall?

We have now further extended the observations, made some new figures, and modified our views and discussion accordingly. The observed along-isopycnal warming in autumn has two possible sources, and it is unlikely to be solely driven by outflow from Ilulissat Icefjord. The relative importance of the two sources is hard to pin down, but we now suggest that the advection from outside the bay is dominating. The new profiles shown in new Fig. 4 (from north of Aasiaat) show this clearly.

It was not our intention to imply that the GMW was the predominant source, just that it was a physically plausible mechanism. We noted a warm signal near the fjord that later appeared at the Monitoring Station, which we suggested could contribute to the autumn warming through advection. However, the 2018 spatial surveys were not concurrent with the 2022-2024 seasonal time series, as the reviewer also pointed out. This and the absence of profiles upstream, inside the fjord, or on the shelf, made it difficult to determine the origin of the along-isopycnal warming with certainty. We therefore suggested GMW as a possible source.

In the revised manuscript, we replace the 2018 dataset with newly available spatial surveys from summers 2022-2024. This includes profiles upstream of Ilulissat Icefjord at the southwestern end of Disko Bay. Accordingly, we revised the methods, results, and discussion. These surveys show that the warm along-isopycnal anomaly was consistently more prominent at the southwestern end of the bay than near Ilulissat. Furthermore, we show that by October-November, hydrographic properties at the Monitoring Station closely matched those measured at the SW entry to Disko Bay in August, approximately two to three months prior (3 months in 2022 and 2024, 2 months in 2023). Moreover, the magnitude of the autumn warming at Monitoring Station along a given isopycnal (e.g. 1.65°C in 2022, 1.1°C in 2023, and 0.48°C in 2024 between August and November at $\sigma_0 = 26.8$ kg m$^{-3}$) is similar to the along-isopycnal temperature difference between Aasiaat and the Monitoring Station in the spatial survey data in August of those same years (1.89°C in 2022, 1.1°C in 2023, and 0.56°C in 2024 ).

This was consistent for 2022, 2023 and 2024 (new Fig. 7) and suggests that the autumn along-isopycnal warming at the Monitoring Station is more likely linked to this signal advected cyclonically around the bay. We updated the discussion to reflect this change and to address

this major comment explicitly. In the updated discussion, we interpret the autumn along-isopycnal warming primarily as the seasonal signal of the WGC, while GMW exported from Ilulissat Icefjord may provide a secondary, but smaller, contribution.

3) The presentation quality is average. I think the figures could be improved for clarity and I mention specific instances in comments below. Also, in all figure captions, it would be helpful to specify what data sources were used, as it was hard to follow along and keep things straight, given there was the 'monitoring' site, the floats, and the 2018 spatial survey all used for different things. Simplifying the results section would help here, as a lot of discussion seems to occur in the results, in addition to the formal discussion section.

In the revised manuscript, we simplified and reorganised the Results section to improve readability and reduce overlap with the Discussion. We also revised all figure captions and legends to state the data sources used clearly.

4) The conclusion that GMW entrain Atlantic-origin water in Greenland tidewater glacier fjords and produce outflows with temperature anomalies relative to their depth is not new. This has been shown over and over again in many different systems (e.g., Straneo et al 2012 looked at several sites around the ice sheet). Many more references since have examined this dynamic.

We agree with the reviewer that the role of Atlantic-origin entrainment in producing warm anomalies in the outflows is well established for Greenland fjords. Our observations were also consistent with this process, so we had, and still have, included it in our discussion. In the revised manuscript, we improved the discussion by providing a clearer context and citing a more comprehensive set of relevant studies, including Straneo et al. (2012) and subsequent work.

5) In the results and abstract they mention that renewal can occur in Fall given upwelling winds. However, it appears like there were consistent upwelling winds in the second year as well, yet no similar renewal occurred. Was there a reason for that? It seems hard to claim that it is part of the seasonal cycle if you have it observed for 1 out of 2 years of data collected, i.e., some other process(es) must be relevant.

In the revised results, we now distinguish more clearly between autumn 2022 and autumn 2023. In autumn 2022, strong and persistent upwelling-favourable winds coincided with a rapid increase in basin density and temperature. In contrast, during autumn 2023 the WGIW boundary shoaled by ~50 m, but no increase in basin density or temperature was observed, indicating that renewal did not occur. We have also made a more quantitative comparison of wind forcing and upwelling velocities between the two autumns, showing that brief episodes of negative wind stress were present in October-November 2023, but weaker and less persistent than in autumn 2022. Vertical velocities were also lower in comparison. While we observe a shoaling of the WGIW boundary in autumn 2023, densities of water over the sill did not exceed those of the basin, and thus basin renewal was not observed.

We agree with the reviewer that observing renewal in only one autumn limits the strength of the seasonal interpretation. Accordingly, we have revised the manuscript to de-emphasise autumn renewal as a regular feature of the seasonal cycle. Instead, we now frame it as a process that can occur under favourable conditions, with variability between years.

**Line by Line Comments**

Line 25: AW can enter shallower than 200-250 m too (as you show later in this paper!), but the sill sets the maximum depth of water that can flow into the icefjord.

Thank you for pointing this out. We clarified the text to avoid implying that AW is confined to these depths.

Line 40: I am a bit worried throughout by the use of a summertime 2018 spatial survey with the time series data from 2022-2024. Summer 2018 was at the end of the cool period noted by other authors. Is there some way to give context for the interannual variations in these data? Maybe the new Picton et al. paper could help.

We appreciate the reviewer's concern. We have also come to fully agree that combining a spatial survey from summer 2018 with the time series from 2022-2024 introduced some inconsistencies. Particularly given that 2018 occurred near the end of a relatively cool period in Disko Bay, as described by Khazendar et al. (2019), Joughin et al. (2020), and more recently by Picton et al. (2025). To address this concern and provide a temporally consistent framework for our analysis, we have revised the manuscript to replace the 2018 spatial survey dataset with newly available spatial surveys from the summers 2022, 2023, and 2024, conducted by the GEM programme.

The change has multiple benefits that will be reflected in the manuscript:

- Temporal consistency: all spatial and time series data now originate from the same period (2022-2024), ensuring that our analysis avoids mixing data from two distinct hydrographic regimes.
- Stronger spatial analysis: the temporal overlap between the new spatial surveys and the time series we focus on in the manuscript enhances our ability to address spatial variability, which was raised in other comments. We refer to the new spatial analysis in response to these comments.
- The updated surveys include new profiles from regions further upstream of Ilulissat Icefjord, which help clarify the origins and pathways of the warm signal discussed throughout the manuscript.

Accordingly, we have revised the Methods section "Spatial hydrographic surveys in 2022-2024" and added a new Results section "Spatial variability".

Line 43: Using these initial guiding questions to structure your discussion might be a way to tie the paper together more. Otherwise, they do not need to be included here. Overall, the tone of the paper is informal and often falls into the trap of (i) telling us what you are about to write about, (ii) telling us, and then (iii) summarizing what you just told us. That lengthens the paper quite a bit. I would try to cut out the preambles on all your results, etc. and just get to the point. One example is at line 79 where you could delete "Below we describe each dataset".

Thank you for your helpful suggestions. In the revised manuscript, we have removed unnecessary openings and adopted a more concise tone. We have restructured the results and discussion to avoid repetition.

Line 59: Is the WGCC relevant to Disko Bay given this uncertainly around whether it even exists at this latitude? That is, does it matter if it's the WGC or WGCC?

We agree with the reviewer that the distinction between the West Greenland Current (WGC) and a possible West Greenland Coastal Current (WGCC) is not well established at the latitude of Disko Bay. In the revised manuscript, we have updated Section 2 to reflect this uncertainty and now refer only to the WGC.

Figure 1: I think you should try to add on the 2018 survey locations to this figure, so all the observations are on one map. They could be small dots. Also, it is impossible to see the crosses used for the float's first profile. Finally, can you label Vaigat Strait here too since it is mentioned a couple times in the text?

This was a helpful suggestion. In the revised manuscript, we have (i) added the 2022–2024 survey station locations to Figure 1, (ii) replaced the float first-profile crosses with larger diamond markers to ensure visibility, and (iii) labelled Vaigat Strait on the map.

Line 103: float data 'were' (data are plural)

Corrected.

Line 110: Not clear what CTD observations the float data were compared to?

This has been clarified, thank you. Salinity obtained from both floats was compared against CTD observations collected at the Monitoring Station (now extended until November 2024). This was done during periods of temporal overlap, as well as against spatial hydrographic surveys (new 2022-2024 data). These comparisons confirmed that the T-S relationship at higher densities exhibits spatial heterogeneity across Disko Bay. Accordingly, float salinities in these density ranges were compared against the full set of available CTD observations in T-S space. Based on this, we found that salinity sensor drift did not exceed 0.02 PSU over the period of data used from either float.

Line 129: Unless the Semper et al 2024 paper is published, it is not helpful to cite here. I would outline the method more completely or maybe show an example or two in a supplement. It's also not stated what threshold you use for the 'sharp increase' in normalized sum-of-squared errors or if the final MLDs are sensitive to this threshold. Finally, it's a two-step method and I clearly see the 'first' step. Is the second step the checking with the 1 standard deviation envelope?

The Semper et al. (2025) paper has now been published, and we cite it directly. Nonetheless, we have expanded the description of our method for determining MLD in the revised manuscript to ensure clarity and reproducibility.

First, for each profile we compute the normalised sum-of-squared errors (SSE) between individual density values and the depth-averaged density, calculated from the surface down to all possible depth intervals. The mixed layer depth (MLD) is identified at the depth where a sharp increase in normalised SSE occurs, indicating a transition from a well-mixed surface layer to more stratified layers below. We applied a threshold of $1.5 \times 10^{-4}$ kg² m⁻⁶ to detect this transition, which we found to be robust across the profiles.

Second, we verify the computed MLD by assessing whether temperature, salinity, and density from the surface to the computed MLD fall within one standard deviation of their respective mean values (Pickart et al., 2002; Semper et al., 2024).

Additionally, we have clarified that in some profiles, no MLD was detected, either because the mixed layer was shallower than the first measured depth or because the surface layer was stratified.

Line 135: The PW layering in Fig 2b is very muted, especially compared to other profiles and systems I've seen published. I assume this is because there are many profiles with surface layers that are not PW? I know that you lump all cool, fresher waters together in this PW layer, but it might be stated more up front before talking about this figure. In addition, the different sources/types of PW presumably have their own seasonality (and in fact some of your discussion of how the mixed layer changes over the year is linked to this).

This was also a helpful comment. We agree that the layering of PW appeared muted in the original version of Fig. 2b and that further clarification improved clarity of the PW definition and the seasonal variability associated with its different sources.

To address these points, we have made the following revisions:

Figure 2 updates:

-   In the updated Fig.2 we changed the (b) panel to show average seasonal profiles of temperature and salinity based on all observations, instead of lumping them together into one average.
-   These seasonal mean profiles more clearly show PW structure, particularly in summer, when the temperature minimum below a warm surface layer becomes more distinct (less muted).
-   This new representation improves the visualisation of PW stratification and its seasonal evolution, as well as makes it possible to show the PW/WGIW layering in more detail than in the previous version of the figure.

Clarification in Methods Section:

-   As suggested, we have added an explicit clarification of PW definition. We now clearly state that we adopt a broad definition of PW, encompassing a range of cool, fresh water types.
-   This clarification acknowledges that each component may vary seasonally, and that the resulting PW structure reflects the cumulative influence of these sources.

Revised Results Section:

In the revised results, we now explicitly integrate the concept, as highlighted by the reviewer, that different components of PW exhibit distinct seasonal behaviours. This includes the development of a stratified surface following sea-ice melt and contributions from other freshwater sources later in the season (see updated Figure 2).

Line 154-155: Is a mixed layer always present? Inside the icefjord and I imagine in Disko Bay at least at times there is a stratified surface layer due to ice melt or other freshwater inputs.

Good question, the answer is no. The presence of a mixed layer is not always observable. To clarify this, we've updated the former Figure 4 and the methods of how we determine MLD. We also added empty markers to this figure to indicate instances where the profile lacked a detectable mixed layer or where the layer was shallower than the first near-surface observation. Consequently, in these cases, the mixed-layer depth, temperature, and salinity are not plotted or calculated. We corrected the manuscript text accordingly.

Line 156-162: I think this paragraph could be shortened, specifically the part about icebergs. Just say they are part of the freshwater flux.

We agree, this is shortened in the revised manuscript.

Line 172: For the wind stresses, you show multiple equations for the drag coefficients, but never the equation for the actual wind stress (which presumably has the drag coefficient in it). Given the relatively small amount of discussion on the Ekman pumping calculation, this section seems long.

Agreed. The wind stress formulation is included, and the section is shortened in the revised manuscript.

Figure 2: Note that iceberg or glacier melt would pull water masses along a line with the same slope as the SMW line, but not necessarily on that exact line shown. That is, the slope is important, but not the exact location and that is true for SGD too, as it matters where the outflowing plume reaches neutral density. All those pink dots in the GMW section seem like they are being pulled down the melt line, but this isn't clear from the figure or the text.

This was also a helpful comment, and we fully agree. It is the slope of the mixing lines between water masses and SGD(runoff)/SMW that carries physical significance in the context of meltwater mixing, and the exact position of these lines depends on local conditions. This point has now been clarified in both the revised figure and the updated manuscript text, as outlined below:

Figure 2 revision:

The original figure (where colour represented depth and all observations were shown in a single panel) has been replaced with a more informative, multi-panel figure:

- Panels a-c (instead of former panel a): Show all T-S observations, colour-coded by season and separated by year (2022, 2023, 2024), highlighting seasonal and interannual variability and more clearly illustrating temporal development.
- Panels d-e (instead of former panel b): Present vertical temperature and salinity profiles, with mean seasonal profiles and clear demarcation between PW and WGIW. This update also responds to the "Line 135:.." suggestion.
- The SMW and SGD mixing lines have been retained, but now are clearly intended as reference slopes, rather than having a specific mixing line with WGIW as before.

The revision in Results :

We have revised the text that could imply that observations follow a specific mixing line between WGIW-SGD or WGIW-SMW. Instead, the new text now describes that some observations align with the slope of SGD (runoff) or SMW mixing lines.

Line 202: 'Both' is wrong, as you list three things here...and then you have a 1 sentence paragraph.

Corrected. Thank you.

Fig 5,6,8: I wonder if these figures could be combined somehow for more visual impact and being able to compare the timing. I think this is where my confusion over the source of the deeper warming comes in, as these figures are all getting at the same thing but are separated.

Good suggestion, and we agree. We have combined the former Figures 5 and 8 into a single figure in the revised manuscript, improving clarity and enabling a direct comparison of timing. To complement this, we expanded and updated the former Figure 6, which remains important as a stand-alone figure presenting distinct results. Specifically, we now show T-S diagrams for the Monitoring station for the summer-autumn periods of 2022, 2023, and 2024. We use these T-S diagrams, overlaid by spatial survey data from August each year, to show that the October-November properties closely resemble those observed upstream two months earlier (addressing the Major comment #2).

Fig 8 and analysis: Float 2 seems to stay relatively stationary but Float 1 moves quite a distance. In Figure 8 and in the results, how much of the 'Float 1, 2022-23' variability is due to time and not a spatial gradient? It's interesting that the second-year variability is lessened given that that float didn't move as much.

In the revised manuscript, we now base our Polar Water analyses solely on the stationary Monitoring Station. We agree that float measurements can alias spatial and temporal variability, the former being particularly pronounced in PW as the revised "Spatial Variability" showed with newly added 2022-2024 hydrographic surveys. Nevertheless, to address this question: Float 2 covered autumns 2023 and 2024, and the reduced variability it recorded (particularly in 2024) is consistent with the new 2024 Monitoring Station data, which also show that the temporal variability of autumn warming was larger in 2022 than in either 2023 or 2024. Thus, while we no longer use floats directly in PW analysis, the larger variability observed by Float 1 still reflected a real temporal signal, not only a spatial artefact.

Line 263: 'representation' of what?

This has been revised. We wanted to state that comparing properties along isopycnals rather than fixed depths provides a more accurate description of the water mass differences across the bay.

Figure 6: The colored dots on the T-S diagram are fairly hard to distinguish. Either try different colors or make them bigger markers.

The new version of Fig. 6 has increased marker size to improve visibility.

Figure 7: Labelling regions on Fig 7a would be helpful.

In the revised manuscript, Figure 7 (2018 spatial analysis) has been removed and replaced with updated spatial analyses from 2022–2024. In all the figures, we have now ensured clear labelling of data sources and regions/locations.

Line 280: But the subglacial discharge that you surmise is causing this is certainly a transient feature, as it ramps up each melt season and dramatically ramps down by end of August or Sept. If the along-isopycnal warming commenced in August, does that imply a 2-month lag time in the icefjord for waters to come out given Float 2's position? Does that jive with estimates of the icefjord circulation/advective scale?

This is an excellent question. As noted in our response to earlier comment ("Fig 8 and analysis: …"), we now base all PW results on the Monitoring Station data. The onset of along-isopycnal warming in August is consistently observed in 2022, 2023, and 2024. As discussed in our earlier response to Major Comment #2, we suggest that this reflects the advection of a warm signal entering the bay from the southwest and propagating cyclonically. We estimated the lag between the SW'rn entry to the bay (station near Aasiaat) and the Monitoring Station is approximately 2-3 months.

Although we cannot resolve the precise advective pathway, synoptic cruise data show consistent T-S differences that support cyclonic circulation past the vicinity of Ilulissat Icefjord. A similar analysis to the one outlined in Major Comment #2, but using a station north of Ilulissat, indicates a comparable relationship in one August survey, but with a shorter (≈1 month) lag. In all years, the Aasiaat station was the warmest in August, with the anomaly weakening progressively along the cyclonic path.

We can make a rough estimate of the mean subsurface velocities:

- The distance between Aasiaat and Monitoring Station of ~170-200km and a 2-month lag implies mean velocities of ~3.3-3.9 cm/s; the same for a 3-month lag implies ~2.2-2.6 cm/s.
- The distance between north of Ilulissat Icefjord and the Monitoring Station of ~90km and a 1-month lag implies mean velocities of ~3.5 cm/s.

To our knowledge, advective pathways and velocities within Disko Bay are not yet well constrained beyond the presence of a cyclonic circulation. Nevertheless, our estimates of ~2.2-3.9 cm/s fall within a plausible range and are comparable to the <10 cm/s velocities at 160m depth recorded in Disko Bay by Sloth and Buch (1984) and ~2-7 cm/s subsurface WGC velocities (Curry et al., 2014). These are also within ranges measured within the top ~200m in Ilulissat Icefjord Gladish (2015, part 1), and mean velocities of ~4 cm/s in Sermilik Fjord (Jackson and Straneo, 2016; Jackson et al., 2018).

Figure 9: It took me a while to infer these data were from the floats (I think?). Indicating that in the caption would be helpful. Also, you mention shading for d-e, but the shaded areas are different. There is also no mention in the text of the large data gap in the second year, which precludes the specification of the start of the renewal.

We updated the former Figure 9 (will be new Figure 8) to include labels and clarified the caption for panels (e-f), where data comes from the Floats (as Monitoring Station observations

don't extend to 400m depths). To avoid confusion with a mismatch of shading, we removed the shading from panel (d) and replaced it with horizontal lines indicating the mean WGIW in the periods between abrupt changes.

In addition, we revised the text to acknowledge two data gaps in the second annual cycle shown in the figure (October 2023-January 2024; January-March 2024). This is because no float observations were deep enough during these periods, now explained clearly. Nevertheless, as density and temperature did not increase between October 2023 and January 2024 we conclude that basin renewal likely did not occur that autumn. For the second gap, between January and March 2024, we now note that density and temperature increased, so the spring renewal has likely begun in that period.

Figure 9 discussion: Again, as for the float data above, I would like some justification for using the float data as time series again given the movement of Float 1. How much variability would you expect over this distance (maybe you can use the 2018 spatial survey to estimate this)?

Thank you for raising this important question. To address it, we conducted a spatial analysis using the 2022-2024 hydrographic surveys and added a new section, "Spatial variability". We analysed spatial variability within PW and WGIW across five near-synoptic cruises with repeated sampling locations (August 2022, May and August 2023, May and August 2024) and included data from Float 1 and 2 that overlapped with these cruises.

For PW properties, we found considerable spatial variability along isopycnals and at constant depths. The revised manuscript will state that "understanding PW seasonality is better achieved using fixed-point observations from the Monitoring Station, as the pronounced spatial variability within PW will interfere with the seasonal patterns inferred from Float data".

In contrast, WGIW exhibits minimal spatial variability along isopycnals. Synoptic observations within the WGIW fall along a narrow line in the T-S space for all five cruises. Some spatial variability is present at constant depths, but it is relatively small and decreases with depth. We will also provide a summary table of this analysis, which is also shown below. Across the five cruises PW/WGIW boundary varied with a standard deviation of 20 to 30m in depth among all sampled locations. Density variability averaged approximately 0.023 kg m$^{-3}$ at 300m and 0.014 kg m$^{-3}$ at 400m depths, based on within-cruise variability in density at each depth.

Given that Floats sampled the deeper parts of the bay, our analysis of WGIW seasonality relies largely on float data. These observations can be used to assess WGIW variability along isopycnals confidently, but also along constant depth levels, provided that the signals exceed the background spatial variability noted above.

We now address your question regarding Float 1 specifically. Its wide spatial coverage between August 2022 and May 2023 is considered acceptable for studying WGIW properties, because the temporal signals this Float captured with respect to WGIW renewal at 400m depth (0.05 kg m$^{-3}$ increase in November-December 2022 and 0.03 kg m$^{-3}$ increase in spring 2023) exceed the background spatial variability observed in the summer cruises of these years.

| Year | Mean WGIW boundary depth (m) | SD of WGIW boundary depth (m) | SD of density at 300 m (kg m$^{-3}$) | SD of density at 400 m (kg m$^{-3}$) |
|---|---|---|---|---|
| 2022 Aug | 250 | 30 | 0.0187 | 0.0132 |
| 2023 May | 156 | 27 | 0.0156 | 0.0148 |
| 2023 Aug | 244 | 24 | 0.0201 | 0.0048 |
| 2024 May | 238 | 23 | 0.0262 | 0.0175 |
| 2024 Aug | 320 | 23 | 0.0343 | 0.0217 |

Moreover, Float 1 moved very little during the spring renewal. As we noted in the methods, between the 6th of February and the 4th of April, 2023, the float was profiling underneath the sea ice, with no known positions. But between the acquisitions with known positions, the float's position changed only by 3.1km. During the autumn renewal in November-December 2022, Float 1 drifted by about 13km. Below is a figure showing positions covered during this time with black markers. Yet, over this time, the WGIW boundary shoaled by >100m, by far exceeding the standard deviation in boundary depth noted in spatial analysis. This demonstrates that the observed renewal reflects a robust signal rather than an artefact of float drift.

[Figure]

Line 319: I don't follow the statement that wind forcing was not evident during this period. It certainly looks like upwelling favourable winds were present at that time?

Thank you for pointing this out. The original statement about wind forcing not being evident during spring 2024 was unclear. Our revised results (strengthened with a more quantitative analysis of winds in response to the second reviewer's suggestion) confirm that upwelling-favourable winds were indeed present during this period, although they were weaker than those observed in spring 2023.

Line 323-334: A lot of this paragraph would fit better into the discussion. There is some redundancy between the results and the discussion, which I think leads to some confusion and adding to the length of the paper.

We moved this into discussion and updated the manuscript to reduce such redundancy throughout.

Line 333: I don't see 1.5C, maybe 1C increase?

Corrected.

Line 359: You cite 'typical' ranges here from previous work, but not anything about the range? That is, is it reasonable for this feature to be advective given variability in these conditions?

At ~150 m in Davis Strait, salinity rarely fell below 33.9 g kg$^{-1}$ (never below 33.8 g kg$^{-1}$), and, unless it was an anomalous year like 2011, density remained above $\sigma_0 = 27$ kg m$^{-3}$ over the 6 years shown in Figure 4 of Gladish et al., (2015 part 2). We were examining the minima to understand the origin of the fresh signal extending over 150m in Disko Bay. Thus, we contrast these numbers to density and salinity we observe at 150m depth in Disko Bay, where minima in autumn is lower ($\sigma_0 \approx 26.9$ kg m$^{-3}$ and $S_A \approx 33.84$ g kg$^{-1}$ Monitoring St. November 2022; $\sigma_0 \approx 26.7$ kg m$^{-3}$ and $S_A \approx 33.59$ g kg$^{-1}$ Monitoring St. October 2023; $\sigma_0 \approx 26.7$ kg m$^{-3}$ and $S_A \approx 33.58$ g kg$^{-1}$ Monitoring St October 2024). Given that these values fall below the Davis Strait minima, we find that local freshwater sources within Disko Bay must also contribute, rather than the feature being fully explained by advection from the WGC.

Line 363: You cite both SGD and SMW are sources of freshwater, which is true. But they have one fundamental difference. For iceberg melt, the depth of entrainment is much much shallower than for SGD, so it's really the SGD that is controlling the depth of the GMW layer (well, and the sill). There is some new literature on refluxing at the icefjord sill (Hager et al. 2024) that might be useful as well to explore. This whole section is relatively long for describing a process that other studies have shown already. It's not really the novel or interesting part of this present study- that's more on the WGIW properties in my opinion.

Thank you for this constructive comment and suggestion on how to improve this section. In the revised manuscript, we have substantially shortened and refocused the discussion of freshwater sources within Ilulissat Icefjord. We now outline the key factors that determine the production and depth of GMW, and address iceberg melt separately, highlighting its role in modifying water mass properties within the fjord. In doing so, we refer to Hager et al. (2024) and other relevant studies, but keep the section concise so that we can clearly summarise the sources of freshwater within the fjord before moving on to discuss the properties of its export to Disko Bay, referring to the study of Beaird et al. 2017.

Line 393: do you mean 'width' here instead of height? As you also have draft? I'm not sure any of those papers actually have measurements of iceberg draft, but assume some geometry based on above water volume and ice/ocean densities.

We agree that our wording was unclear, and this has been revised. We now explicitly describe the assumed iceberg geometry: starting from the "small iceberg" with an area of ~1800 m$^2$ (Scheick et al., 2019), then representing such iceberg as 130m in length and width, from which we estimate it is 65m thick (freeboard+draft, following Enderlin et al., 2016 and their Figure 3a), and has a draft of 55m based on a freeboard-to-draft ratio of 1:7 (Cenedese and Straneo, 2023). We have also replaced "height" with "thickness" to avoid confusion.

Section 5.3: I am not sure this section says very much. A lot of it is speculation about how the results might be important to marine ecosystems. The last paragraph in particular is vague. Adding some meat/content would be good. For example, can you show the cyclonic circulation sense in the 2018 spatial survey (not just in T/S but in dynamic sections)?

We appreciate this comment and agree that Section 5.2 required improvement. In the revised manuscript, we have not included the 2018 data and therefore do not attempt to infer circulation from those sections. Instead, we have revised this section to reduce speculation and to more directly highlight the relevance of our findings. Specifically, we connect the observed seasonality to existing ecosystem studies in Disko Bay, particularly those focused on the Monitoring Station. By grounding the section in ongoing research, we aim to show that a better-resolved oceanographic seasonality provides a useful framework for interpreting ecosystem variability, while avoiding unsupported speculation.

Line 446: 'de-seasoning' is not a word.

Corrected.

Line 453: This sentence and the one before it are confusing. They say nutrients are entrained deep in the icefjord, flow out into Disko Bay and then into the icefjord? Maybe I'm missing something here.

The text has been revised for clarity.

General comment: Is it possible to update the schematic or create your own schematic (Fig 15 of Gladish et al (part 2))? This might help the reader understand what is new here or what is validating previous ideas (which is important too). I think the data here are very cool!

Thank you again for your positive and constructive suggestions. We have now created an update to the schematic Figure 15 by Gladish et (2015). Our results provided observational evidence for the mechanism they propose in Spring/Summer (lower panel of their figure), which was well-represented in their schematic. We updated the upper panel to include the possible wind-driven renewal mechanism in Autumn, which was not considered. This will be included as a new figure in the revised manuscript.

---

## Author Comment (AC2)

**Response to Reviewer#2**

We sincerely thank you for your positive and constructive review of our manuscript. Your feedback helped us to improve the clarity and readability of the manuscript. We hope the addition of a new schematic figure and a more quantitative analysis of wind forcing, as you recommended, will assist readers in understanding the key mechanisms of WGIW renewal more clearly.

Please find our detailed responses in blue below. Most of your comments have been fully addressed. In addition, several further improvements were made while addressing Reviewer 1's comments, including incorporation of spatial surveys from 2022-2024, extension of Monitoring Station data until November 2024, and a more concise and structured presentation of our results and discussion. These changes have refined our interpretation of the source of the observed autumn along-isopycnal warming, whilst ensuring that the results on WGIW renewal remain central to the paper.

Best regards,

Linda Latuta on behalf of all co-authors.

**Reviewer#2 Summary**

This manuscript by Latuta et al. presents analyses of hydrographic dataset in Disko Bay, Greenland. It provides insights into the seasonality and spatial variability of the water masses in the bay system and gives explanations on the potential physical mechanisms that drive the seasonal cycles. Overall, I believe the manuscript would contribute to further our understanding of the hydrography in this area and the finding are significant and timely. But I also think the presentations could be improved and some clarifications should be made before acceptance.

**General comments**

1. A key explanation shown in this work is the identification of wind-driven WGIW renewal via upwelling-favorable conditions at the EDS. However, this mechanism (e.g., around line 305) is complex and may be difficult to visualize for some readers. I recommend including a schematic figure showing the seasonal wind changes, upwelling over EDS and the resultant basin renewal.

   We appreciate this valuable suggestion. To improve clarity, we have added a new schematic figure illustrating the seasonal mechanisms of WGIW renewal. The figure builds on the schematic of Gladish et al. (2015, part 2, Figure 15) but is adapted to highlight the processes identified in our study. In particular, it shows wind-driven upwelling over the Egedesminde Dyb sill and the resulting basin renewal observed under favourable wind forcing in autumn 2022. We believe this schematic will help readers to visualise the dynamics better.

2. The manuscript suggests that upwelling-favorable winds caused WGIW renewal in November and December 2022, but the upwelling effect was not as effective during autumn and winter 2023-2024 (50m uplift; line 317). It would be helpful to provide an estimate of the threshold of the wind stress or its duration needed to cause upwelling

sufficient to lift WGIW over EDS. Is there any lag between wind stress and the dense water renewal? Could the authors quantify this relationship?

Thank you for this helpful suggestion. In the revised manuscript, we now quantify the relationship between wind forcing and calculated vertical velocities. Based on daily means, we find that "strong vertical velocities" (upper quartile of distribution) are characterised by velocities greater than ~1m day$^{-1}$. Such events typically lasted several days but often occurred in clusters extending over 2-3 weeks, associated with sustained upwelling-favourable winds with wind stress exceeding $\leq -0.1$ N m$^{-2}$. Cross-correlation analysis indicated that the Ekman response follows negative wind stress within hours, as expected given its derivation from wind stress (Eq. 5). Using 10-day means to emphasise persistent conditions (as plotted in Figure 8), we find that vertical velocities of ~0.35–0.45 m day$^{-1}$ were attained under an average wind stress $\leq -0.055$ N m$^{-2}$ sustained over multiple days to weeks. We also show that in autumn 2023, the wind stress was weaker and less persistent than these thresholds.

When relating upwelling to WGIW boundary depth, we found only a weak statistical relationship. This is expected for several reasons, and we note that quantifying this relationship with the available observations is complicated. First, our vertical velocities are calculated from wind stress and represent the potential Ekman pumping over the EDS. However, we lack hydrographic observations west of or over EDS to observe the actual upwelling, determine the actual density of the uplifted waters, the magnitude of that vertical displacement, and the extent to which other processes, such as coastal upwelling, may have contributed. Second, the vertical uplift at the sill is not necessarily proportional to uplift within the Disko Bay basin below sill depth. Third, the WGIW boundary depth time series combines data from two Floats and Monitoring Station data. These represent different sampling locations, and the WGIW boundary depth itself varies by ~30m across Disko Bay. We will include a new Table 2, along with other results from our newly improved spatial analysis. This inherent spatial variability limits our ability to define a robust quantitative relationship between wind forcing and WGIW boundary depth. Finally, the advective timescales between the inflow of dense waters across EDS and their arrival at the observation sites are not well constrained.

3. The sampling frequency in Table 1 seems a bit long to capture variability in shorter time scales. It would be great if the authors could discuss whether the variability in higher frequencies would affect the water mass exchanges in the bay system.

We appreciate this comment and suggestion. While the sampling frequency of our data may appear sparse, it is unique in that it includes late autumn, winter, and early spring observations (repeated over two annual cycles). These periods are rarely sampled in Disko Bay. The Monitoring Station profiles (weekly-monthly, repeated three years, with a targeted autumn campaign in 2023 to enhance observational frequency) combined with the floats sampling at 5-day intervals provide the longest and highest-frequency seasonal timeseries available. Together, these complementary datasets resolve the seasonal processes we focus on, with signals that are persistent and repeated from year to year.

We agree that higher frequency variability is certainly present, but in the context of seasonal forcing, we believe that the main physical processes are well captured. As part of the revision (see response to Reviewer 1), we have substantially shortened the manuscript and therefore, respectfully, choose not to add an additional discussion about higher-frequency variability, which lies beyond the scope of this study.

**Specific comments**

Figure 1 and lines 61-70: I notice the lack of labels for locations in the diagram, and it was difficult to follow the statement without further searching for another map online. Please include some key locations such as Vaigat Strait to help the reader navigate.

We have revised Figure 1 with the suggestions implemented.

Line 165 and Figure 2 caption: I realize the mixing line definition is not explicitly indicated in the Figure 2 caption. It would be more straightforward if the reader could find the definition of the mixing line end-points without referring back to the T-S statement in the manuscript.

Thank you for this helpful suggestion. We revised the figure caption accordingly.

Figure 7a: please label the boxes with eastern, northern, and central.

In the revised manuscript, Figure 7 (2018 spatial analysis) has been removed and replaced with updated spatial analyses from 2022–2024. In all the updated figures, we have now ensured clear labelling of regions/locations.

Line 319: it seems that in March and April 2024, the wind forcing and upwelling were strong (the same period in Figure 9c as the orange shading in Figure 9ef). Why does the statement here say "not evident"?

Thank you for pointing this out. The original statement that wind forcing was not evident during spring 2024 was unclear. Our revised results (strengthened with a more quantitative analysis of winds following the general comment #2) confirm that upwelling-favourable winds were indeed present during this period, although they were weaker than those observed in spring 2023.

Figure 10: please label the panels (a) and (b).

We labelled the panels in the former Figure 10 (now Figure 9 in the revised manuscript).

---

## Author Response (AR1)

**Response to Reviewer#1**

We thank you for your constructive and thoughtful review of our manuscript. Your suggestions have been invaluable in improving the quality and clarity of our work. In particular, your feedback prompted several substantial revisions: (1) replacing the 2018 dataset with newly available spatial surveys from summers 2022-2024, (2) conducting a thorough spatial analysis based on these new surveys, and (3) incorporating the newly available Monitoring Station observations from 2024. These revisions were central to addressing your major comments and have considerably strengthened the manuscript.

Please find our detailed responses in blue below. In the few cases where it was not possible to follow your suggestions directly, we provide a detailed explanation in our responses.

Best regards,

Linda Latuta on behalf of all co-authors.

**Reviewer#1 Summary**

This paper presents observational hydrographic data from Disko Bay, Greenland, which sits near a large, fast tidewater glacier draining the Greenland Ice Sheet (Sermeq Kujalleq). They combine existing data sources from a nearby monitoring station (GEM site), two profiling floats, and re-analysis atmospheric & satellite data over two seasonal cycles (the years 2022-2024) along with weekly repeat stations taken over Fall 2023. In the end, they describe a seasonal cycle that is consistent over both years, showing both the modulation of the mixed layer and when the warmest and densest Atlantic-origin waters arrive at the mouth of Ilulissat Icefjord (which leads to Sermeq Kujalleq, SK hereafter). Overall, they present a detailed account of the hydrography in the bay's water mass layers, which should be beneficial to future studies looking to oceanic change as a driver for mass loss from Greenland. Although many pieces of the study are already known, I find the most novel results to be their discussion of the deep-water renewal and lifting of the Atlantic-origin waters above sill depth at the mouth of the icefjord. However, the paper seems overly complicated for the conclusions it reaches, and the figures could be clarified and improved given all the disparate sources of data. For example, explicitly stating what is novel here and what is validation of previous conceptual models of the Bay's exchange with the coast and the icefjord would be helpful. One way to do that would be to update or revise the schematic from Gladish et al. part 2 (his Fig. 15). But there are other ways too. I was also not convinced of the Fall along-isopycnal warming signal they attribute to outflowing glacially modified waters (GMW). Finally, a new paper (Picton et al. 2025) came out in JGR-Earth Surface recently that uses some of the same data (floats and GEM site), although their focus is slightly different.

More on these comments below, as well as line by line comments. I certainly believe these data and this write-up to be publication worthy, but their impact will be significantly improved through revisions.

**Major comments:**

1) During the discussion period, another paper came out that studies the same region with some of the same data sources (Picton et al. 2025). It would be good to include this

reference in the revision. For example, that paper shows temperature vs time at 240 m depth and seems to show a similar Fall warming and seasonality as described in this current study. I think discussion of this new publication will only strengthen this one. In terms of other references, I believe the authors could certainly expand their sources. I point to a few examples in the comments below.

Response:

We appreciate this helpful suggestion, and we also believe the new study by Picton et al. (2025) provides valuable context for our study and strengthens both the Introduction and Discussion. In the revised manuscript, we have incorporated this reference and expanded our literature base to provide a more comprehensive context for our findings.

Changes in the manuscript:

Addition of the following references: Beaird et al. (2015); Caroll et al. (2016, 2017); Cowton et al. (2015); Hager et al. (2024); Hansen et al. (2012); Jackson et al.(2017); Jenkins (2011); Joughin et al. (2004, 2018); Lindeman et al. (2024); Mankoff et al. (2016); Moon et al. (2018); Mortensen et al. (2011, 2014, 2020); Pacini and Pickart (2023); Picton et al (2025); Ribergaard et al (2004); Sanchez et al. (2024); Semper et al. (2024); Slater et al. (2022); Snow et al. (2023); Stevens et al. (2016); Söderkvist et al. (2006); Straneo et al. (2011); Wood et al. (2021, 2025); Wood et al. (2025)

2) In terms of the along-isopycnal warming trend they document and discuss, I am not convinced it's entirely from the outflow from the icefjord. I think what they are saying is that the summer outflow signal is delayed and lagged as it transits Disko Bay, so the warming they see through the Fall is from that entrained WGIW. However, the melt season at the glacier (e.g., look at a subglacial discharge time series for SK) is relatively short and the warming seems to continue beyond the length of a typical melt season. Although they calculate advective time scales for other processes in the paper, they don't estimate the time scales here. Are there other mechanisms that could cause this warming trend through the Fall?

Response:

We thank the reviewer for this important comment, which prompted us to revisit our interpretation of the autumn along-isopycnal warming. In the revised manuscript, we have extended the observational record, made new figures, and substantially revised our results and discussion in relation to the autumn along-isopycnal warming.

The observed warming in autumn has two possible sources, and it is unlikely to be solely driven by outflow from Ilulissat Icefjord. Newly available hydrographic surveys from 2022-2024 (replacing the 2018 dataset) show that the warm anomaly was consistently more pronounced at the southwestern entry to Disko Bay than near Ilulissat Icefjord. Moreover, the autumn hydrography at the Monitoring Station closely matched that measured at the southwestern entry two to three months earlier (3 months in 2022 and 2024, 2 months in 2023). The magnitude of the autumn warming at Monitoring Station along a given isopycnal (e.g. 1.65°C in 2022, 1.1°C in 2023, and 0.48°C in 2024 between August and November at $\sigma_0$ = 26.8 kg m$^{-3}$) is similar to the along-isopycnal temperature difference between Aasiaat and the Monitoring Station in the spatial survey data in August of those same years (1.89°C in 2022, 1.1°C in 2023, and 0.56°C in 2024 ). These results, now presented in new Fig. 7, suggest that  the autumn along-isopycnal

warming at the Monitoring Station is more likely linked to a signal advected cyclonically around the bay. In the updated discussion, we interpret this primarily as the seasonal signal of the WGC, while GMW exported from Ilulissat Icefjord may provide a secondary, but smaller, contribution.

It was not our intention to imply that the GMW was the predominant source. Rather, we suggested it as a plausible driver because a warm signal near the fjord in 2018 data coincided with warming later observed at the Monitoring Station. However, the 2018 spatial surveys were not concurrent with the 2022-2024 seasonal time series, as you also pointed out. This and the absence of profiles upstream, inside the fjord, or on the shelf, made it difficult to determine the origin of the along-isopycnal warming with certainty. With the new dataset and analysis, we believe, our results and interpretation has strengthened.

Changes in the manuscript:

Methods/Data: Replaced 2018 hydrographic survey data with 2022-2024 surveys (updated Section 3.1.3). Extended Monitoring Station timeseries to include data from 2024 (updated Section 3.1.1; Line 78, first row of Table 1).

Results: New section 4.2 ("Spatial variability"); revised Section 4.4 (Lines 249-267); added new Fig. 7, which (1) extends and improves the previous Fig. 6 and (2) replaces the previous analysis and Fig. 7 that were based on the removed 2018 data.

Discussion: Revised Section 5.1.2 ("Along-isopycnal warming at depth") to reflect this new interpretation and explicitly address the reviewer's comment (Lines 372-392)

Conclusion and abstract: Updated accordingly.

3) The presentation quality is average. I think the figures could be improved for clarity and I mention specific instances in comments below. Also, in all figure captions, it would be helpful to specify what data sources were used, as it was hard to follow along and keep things straight, given there was the 'monitoring' site, the floats, and the 2018 spatial survey all used for different things. Simplifying the results section would help here, as a lot of discussion seems to occur in the results, in addition to the formal discussion section.

Response and changes in the manuscript:

In the revised manuscript, we simplified and reorganised the Results section to improve readability and reduce overlap with the Discussion. We also revised all figure captions and legends to state the data sources used clearly.

4) The conclusion that GMW entrain Atlantic-origin water in Greenland tidewater glacier fjords and produce outflows with temperature anomalies relative to their depth is not new. This has been shown over and over again in many different systems (e.g., Straneo et al 2012 looked at several sites around the ice sheet). Many more references since have examined this dynamic.

Response and changes in the manuscript:

We agree with the reviewer and in the revised manuscript, and particularly with the addition of a new spatial survey and a reduced emphasis on GMW (see response to major comment #2),

we improved the discussion by citing a more comprehensive set of relevant studies, including Straneo et al. (2012) and subsequent publications.

5) In the results and abstract they mention that renewal can occur in Fall given upwelling winds. However, it appears like there were consistent upwelling winds in the second year as well, yet no similar renewal occurred. Was there a reason for that? It seems hard to claim that it is part of the seasonal cycle if you have it observed for 1 out of 2 years of data collected, i.e., some other process(es) must be relevant.

Response:

In the revised results, we now distinguish more clearly between autumn 2022 and autumn 2023. In autumn 2022, strong and persistent upwelling-favourable winds coincided with a rapid increase in basin density and temperature, consistent with renewal. In autumn 2023, although the WGIW boundary shoaled by ~50 m, basin density or temperature did not increase. We have also included a more quantitative comparison of wind forcing and upwelling velocities between the two autumns, showing that brief episodes of negative wind stress occurred in October-November 2023, but they were weaker and less persistent than in autumn 2022. Vertical velocities were also lower in comparison. While we observe a shoaling of the WGIW boundary in autumn 2023, densities of water over the sill might not have exceeded those of the basin, and thus, basin renewal was not observed.

We agree that observing renewal in only one autumn limits the strength of the seasonal interpretation. Accordingly, we have revised the manuscript to de-emphasise autumn renewal as a regular seasonal feature and instead present it as a process that can occur under favourable conditions, with interannual variability.

Changes in the manuscript:

Results: Added more detailed analysis of wind forcing in Section 4.5 (Lines 281-306), with direct comparison between autumn 2022 (Lines 281-284) and autumn 2023 (Lines 296-299)

Discussion, conclusion and abstract: Updated to reflect that autumn renewal is not a regular seasonal occurrence (e.g. Lines 11-12, 409-412, 426-431, 451-452)

**Line by Line Comments**

Line 25: AW can enter shallower than 200-250 m too (as you show later in this paper!), but the sill sets the maximum depth of water that can flow into the icefjord.

Response and changes in the manuscript:

Thank you for pointing this out. We clarified the text in Lines 26-32 to avoid implying that AW is confined to these depths.

Line 40: I am a bit worried throughout by the use of a summertime 2018 spatial survey with the time series data from 2022-2024. Summer 2018 was at the end of the cool period noted by other authors. Is there some way to give context for the interannual variations in these data? Maybe the new Picton et al. paper could help.

Response:

We appreciate the reviewer's concern. We agree that combining the 2018 spatial survey with the 2022-2024 time series introduced inconsistencies, particularly since 2018 occurred near the end of a relatively cool period in Disko Bay ( Khazendar et al., 2019;  Joughin et al., 2020; Picton et al., 2025). To address this concern and provide a temporally consistent framework for our analysis, we have revised the manuscript to replace the 2018 spatial survey dataset with newly available spatial surveys from the summers 2022-2024.

The change has multiple benefits that are now reflected in the manuscript:

- Temporal consistency: all spatial and time series data now originate from the same period (2022-2024), ensuring that our analysis avoids mixing data from two distinct hydrographic regimes.
- Stronger spatial analysis: the temporal overlap between the new spatial surveys and the time series we focus on in the manuscript enhances our ability to address spatial variability, which was raised in other comments. We refer to the new spatial analysis in response to these comments.
- The updated surveys include new profiles from regions further upstream of Ilulissat Icefjord, which help clarify the origins and pathways of the warm signal discussed throughout the manuscript.

Changes in the manuscript:

Methods: Replaced 2018 hydrographic surveys with 2022-2024 hydrographic surveys (Section 3.1.3); updated Fig. 1 and Table 1.

Results: Added section 4.2 ("Spatial variability"), revised section 4.4 (Lines 245-267); new Fig. 4 (replaces previous spatial analysis in former Fig. 7) and new Fig. 7 (extends and improves the former Fig. 6)

Line 43: Using these initial guiding questions to structure your discussion might be a way to tie the paper together more. Otherwise, they do not need to be included here. Overall, the tone of the paper is informal and often falls into the trap of (i) telling us what you are about to write about, (ii) telling us, and then (iii) summarizing what you just told us. That lengthens the paper quite a bit. I would try to cut out the preambles on all your results, etc. and just get to the point. One example is at line 79 where you could delete "Below we describe each dataset".

Response:

Thank you for your helpful suggestions. In the revised manuscript, we removed unnecessary openings and adopted a more concise tone throughout. We also restructured the results and discussion to avoid repetition and improve flow.

Changes in the manuscript:

- Tightened language throughout the manuscript to avoid repetition and informal phrasing
- Shortened Section 3.1 by moving information about instrument accuracies to Table 1
- Re-organised Section 4.3 for a more streamlined presentation

- Shortened Section 3.3.3 by citing Lupkes and Brinbaum (2005) parameterisation directly
- Combined previous Sections 4.3.1 and 4.3.2 into one single Section 4.4 ("Seasonality of Polar Water"), with a more streamlined presentation

Line 59: Is the WGCC relevant to Disko Bay given this uncertainly around whether it even exists at this latitude? That is, does it matter if it's the WGC or WGCC?

Response:

We agree with the reviewer that the distinction between the West Greenland Current (WGC) and a possible West Greenland Coastal Current (WGCC) is not well established at the latitude of Disko Bay.

Changes in the manuscript:

Updated Section 2, now referring only to the WGC.

Figure 1: I think you should try to add on the 2018 survey locations to this figure, so all the observations are on one map. They could be small dots. Also, it is impossible to see the crosses used for the float's first profile. Finally, can you label Vaigat Strait here too since it is mentioned a couple times in the text?

Response: Thank you for these helpful suggestions

Changes in the manuscript:

We added the 2022–2024 survey station locations to Figure 1, replaced the float first-profile crosses with larger diamond markers to ensure visibility, and labelled the Vaigat Strait on the map.

Line 103: float data 'were' (data are plural)

Corrected.

Line 110: Not clear what CTD observations the float data were compared to?

Response:

This has been clarified, thank you. Float salinities were compared against CTD observations collected at the Monitoring Station (now extended until November 2024) during periods of temporal overlap, as well as against spatial hydrographic surveys from 2022-2024. These comparisons confirmed that the T-S relationship at higher densities exhibits spatial heterogeneity across Disko Bay. Accordingly, float salinities in these density ranges were compared against the full set of available CTD observations in T-S space. Based on this, we found that salinity sensor drift did not exceed 0.02 PSU over the period of data used from either float.

Changes in the manuscript:

Clarified the float salinity comparison procedure in Lines 102-107.

Line 129: Unless the Semper et al 2024 paper is published, it is not helpful to cite here. I would outline the method more completely or maybe show an example or two in a supplement. It's also not stated what threshold you use for the 'sharp increase' in normalized sum-of-squared errors or if the final MLDs are sensitive to this threshold. Finally, it's a two-step method and I clearly see the 'first' step. Is the second step the checking with the 1 standard deviation envelope?

Response:

The Semper et al. (2024) paper has now been published, and we cite it directly. In addition, we expanded the description of how we determine MLD to ensure clarity and reproducibility. We also have clarified that in some profiles, no MLD was detected, either because the mixed layer was shallower than the first measured depth or because the surface layer was stratified.

Changes in the manuscript:

Expanded and clarified section 3.3.1

Line 135: The PW layering in Fig 2b is very muted, especially compared to other profiles and systems I've seen published. I assume this is because there are many profiles with surface layers that are not PW? I know that you lump all cool, fresher waters together in this PW layer, but it might be stated more up front before talking about this figure. In addition, the different sources/types of PW presumably have their own seasonality (and in fact some of your discussion of how the mixed layer changes over the year is linked to this).

Response:

This was also helpful comment, thank you. We agree that the PW layering in the original version of Fig. 2b appeared muted and that further clarification of the PW definition and a seasonal variability of its sources was needed.

Changes in the manuscript:

Former Fig. 2 (now Fig. 3) revised:

- Former Fig.2b replaced with seasonal mean profiles of temperature and salinity (now Fig.3 d-e), which show PW layering more distinctly, especially in summer.
- Fig. 3a-c (former Fig. 2a) updated to show three annual cycles separately and incorporating the edits made in response to the reviewer's comment: "Figure 2: Note that iceberg or glacier melt would pull water masses…".

Methods (section 3.3.2, Lines 145-154): Added an explicit clarification of the PW definition, noting that different components may vary seasonally and that the resulting PW structure reflects their cumulative influence.

Results (section 4.1, Lines 186-188): Now explicitly present that different PW components exhibit distinct seasonal behaviours.

Line 154-155: Is a mixed layer always present? Inside the icefjord and I imagine in Disko Bay at least at times there is a stratified surface layer due to ice melt or other freshwater inputs.

Response:

Good question. The presence of a mixed layer is not always observable. To clarify this, we updated the former Fig. 4 (now Fig.5) and the methods of determining the MLD. We added empty markers to Fig.5a to indicate instances where the profile lacked a detectable mixed layer or where the layer was shallower than the first near-surface observation. In these cases, the MLD, mixed-layer temperature, and mixed-layer salinity are not plotted or calculated.

Changes in the manuscript:

Methods: Updated Section 3.3.1, Lines 130-131

Results: Fig.5 (former Fig.4) now shows empty circle markers in (a) to indicate times when the mixed layer was not detectable. Lines 227-229 reflect that in-text.

Line 156-162: I think this paragraph could be shortened, specifically the part about icebergs. Just say they are part of the freshwater flux.

Response: We agree. The paragraph has been shortened in the revised manuscript.

Changes in the manuscript: paragraphs in former Lines 156-170 are now shortened and reorganised in new Lines 155-164.

Line 172: For the wind stresses, you show multiple equations for the drag coefficients, but never the equation for the actual wind stress (which presumably has the drag coefficient in it). Given the relatively small amount of discussion on the Ekman pumping calculation, this section seems long.

Response: Thank you for pointing this out. We added an explicit wind stress formulation and streamlined the section to make it more concise.

Changes in the manuscript:

Included the wind stress formulation (new Eq. 1) and shortened the section by removing the previous Eq. 1-4 and referring directly to Lupkes and Brinbaum (2005) for the drag coefficient parameterisation (Lines 170-172).

Figure 2: Note that iceberg or glacier melt would pull water masses along a line with the same slope as the SMW line, but not necessarily on that exact line shown. That is, the slope is important, but not the exact location and that is true for SGD too, as it matters where the outflowing plume reaches neutral density. All those pink dots in the GMW section seem like they are being pulled down the melt line, but this isn't clear from the figure or the text.

Response:

Thank you for this helpful comment, and we are entirely in agreement. This point has now been clarified in both the revised figure and the updated manuscript text.

Changes in the manuscript:

Previous Fig. 2 (now Fig. 3) updates:

- Panels a-c (instead of former panel a): Show all T-S observations, colour-coded by season and separated by year (2022, 2023, 2024), to separate seasonal and interannual variability and better illustrate temporal development.
- Panels d-e (instead of former panel b): Present vertical temperature and salinity profiles, with mean seasonal profiles and clear demarcation between PW and WGIW. This update also addresses the "Line 135:.." reviewer's comment.
- The SMW and SGD mixing are retained only as reference slopes, rather than implying a specific mixing line with WGIW as before.

Results: Text is revised to clarify that observations do not follow a specific mixing line between WGIW-SGD or WGIW-SMW. Instead, the new text (Lines 186-190) now describes that some observations align with the slope of SGD (runoff) or SMW mixing lines.

Discussion (Lines 350-355): Updated accordingly to reflect this clarification.

Line 202: 'Both' is wrong, as you list three things here...and then you have a 1 sentence paragraph.

Response: Corrected. Thank you.

Fig 5,6,8: I wonder if these figures could be combined somehow for more visual impact and being able to compare the timing. I think this is where my confusion over the source of the deeper warming comes in, as these figures are all getting at the same thing but are separated.

Response: Excellent suggestion. Thank you.

Changes in the manuscript:

New Fig. 6 (combines former Fig. 5 and Fig.8):

- Adjusted the depth range over which mean salinity was calculated, from the former upper 150 m to 120 m (panel (a) and results Lines 238-241). This was done to reflect the use of the Monitoring Station alone in the results section 4.4 (as prompted by the reviewer's comment "Fig 8 and analysis…" and our new results section 4.2, Lines 214-215), where $26.5$ kg m$^{-3}$ does not extend below 120 m depth.
- We only highlight Monitoring Station observations (stars), and show Float 1 and 2 observations as faint background markers for reference only.
- We show temperature along $26.2$ kg m$^{-3}$ in panel (b) (within the stratified layer $\sigma_0 < 26.5$ kg m$^{-3}$ ) and $26.8$ kg m$^{-3}$ in panel (c) (below the stratified layer) to avoid redundancy in showing the temperature along two isopycnals below the stratified layer (former Fig. 8a-b).

- Omitted showing the depth of 26.5 kg m$^{-3}$ as in former Fig.5(b), as this can be seen from Fig. 2e, which we refer to directly in Lines 232-234.

New Fig. 7 (former Fig. 6):

- Retained as a stand-alone figure, as it presents distinct results. Expanded to show T-S diagrams for the Monitoring station for the summer-autumn periods of 2022, 2023, and 2024 (previously only 2023).
- Overlaid these with August spatial survey data, showing that the October-November Monitoring Station properties closely resemble upstream conditions seen two to three months earlier (addressing the Major comment #2).

Results and Discussion: Revisions in results section 4.4 and the discussion sections 5.1.1 and 5.1.2 reflect these changes.

Fig 8 and analysis: Float 2 seems to stay relatively stationary but Float 1 moves quite a distance. In Figure 8 and in the results, how much of the 'Float 1, 2022-23' variability is due to time and not a spatial gradient? It's interesting that the second-year variability is lessened given that that float didn't move as much.

Response:

We agree with the reviewer that float measurements can alias spatial and temporal variability, particularly for PW, as also demonstrated in the new Section "Spatial Variability" using 2022–2024 hydrographic surveys. To avoid this ambiguity, our revised PW analyses are now based solely on the stationary Monitoring Station.

To address this specific question: Float 2 covered autumns 2023 and 2024, and the reduced variability it recorded (particularly in 2024) is consistent with the new 2024 Monitoring Station data. These also show that the autumn warming was larger in 2022 than in either 2023 or 2024 (Lines 259-261 in the revised manuscript). Thus, while we no longer use floats directly in PW analysis, the higher variability observed by Float 1 still reflected a real temporal signal, rather than only a spatial artefact.

Line 263: 'representation' of what?

Response:

This text has been removed, as it referred to the former results based on 2018 spatial surveys. These have now been replaced with 2022-2024 spatial surveys.

Figure 6: The colored dots on the T-S diagram are fairly hard to distinguish. Either try different colors or make them bigger markers.

Response: We implemented this suggestion. Thank you.

Changes in the manuscript: Increased marker size in new Fig.7 (former Fig. 6) to improve visibility. Added black outlines to October-November observations to highlight these months, which are frequently referred to in the results.

Figure 7: Labelling regions on Fig 7a would be helpful.

Response:

In the revised manuscript, the former Fig. 7 (2018 spatial analysis) has been removed, as we now use the 2022-2024 spatial surveys instead.

Changes in the manuscript:

All relevant figures now include clear labelling of data sources and regions/locations. For example, towns mentioned in the text are marked in the new Fig. 4 based on 2022-2024 spatial surveys.

Line 280: But the subglacial discharge that you surmise is causing this is certainly a transient feature, as it ramps up each melt season and dramatically ramps down by end of August or Sept. If the along-isopycnal warming commenced in August, does that imply a 2-month lag time in the icefjord for waters to come out given Float 2's position? Does that jive with estimates of the icefjord circulation/advective scale?

Response:

This is an excellent question. As noted in our response to earlier comment ("Fig 8 and analysis: …"), we now base all PW analyses on the Monitoring Station data. The onset of along-isopycnal warming in August is consistently observed in 2022, 2023, and 2024. Following our new analysis (see response to Major Comment #2), we interpret this warming as advection of warm waters entering Disko Bay from the southwest and propagating cyclonically. We estimated that the lag between the SW'rn entry point (station near Aasiaat) and the Monitoring Station is approximately 2-3 months.

Although the exact advective pathway is uncertain, synoptic cruise data show consistent T-S differences that support cyclonic circulation past the vicinity of Ilulissat Icefjord. A similar analysis to the one outlined in response to Major Comment #2, but using a station north of Ilulissat, indicates a comparable relationship in one August survey, but with a shorter (≈1 month) lag. In all years, the Aasiaat station was the warmest in August, with the anomaly weakening progressively along the cyclonic path.

We can make a rough estimate of the mean subsurface velocities:

- The distance between Aasiaat and Monitoring Station of ~170-200km and a 2-month lag implies mean velocities of ~3.3-3.9 cm/s; the same for a 3-month lag implies ~2.2-2.6 cm/s.
- The distance between north of Ilulissat Icefjord and the Monitoring Station of ~90km and a 1-month lag implies mean velocities of ~3.5 cm/s.

To our knowledge, advective pathways and velocities within Disko Bay are not yet well constrained beyond the presence of a cyclonic circulation. Nevertheless, our estimates of ~2.2-

3.9 cm/s fall within a plausible range and are comparable to the <10 cm/s velocities at 160m depth recorded in Disko Bay by Sloth and Buch (1984) and ~2-7 cm/s subsurface WGC velocities (Curry et al., 2014). These are also within ranges measured within the top ~200m in Ilulissat Icefjord Gladish (2015, part 1), and mean velocities of ~4 cm/s in Sermilik Fjord (Jackson and Straneo, 2016; Jackson et al., 2018).

Changes in the manuscript:

Results: Lines 250-267 implement the above response.

Discussion: Revised Section 5.1.2 implements the above response

Conclusion: Lines 444-447 implement the above response.

Figure 9: It took me a while to infer these data were from the floats (I think?). Indicating that in the caption would be helpful. Also, you mention shading for d-e, but the shaded areas are different. There is also no mention in the text of the large data gap in the second year, which precludes the specification of the start of the renewal.

Response: The revised figure and the text now clarify the data sources and acknowledge data gaps.

Changes in the manuscript:

Fig. 8 updates (former Fig.9):

- Included labels and clarified the caption for panels (e-f), where data comes from the Floats (as Monitoring Station observations don't extend to 400m depths).
- To avoid confusion with a mismatch of shading, we removed the shading from panel (d) and replaced it with horizontal lines indicating the mean WGIW boundary depth in the periods between abrupt changes.

Results: The text revised to acknowledge two data gaps in the second annual cycle: October 2023-January 2024 (Line 294-296) and January-March 2024 (Lines 301-303)

Figure 9 discussion: Again, as for the float data above, I would like some justification for using the float data as time series again given the movement of Float 1. How much variability would you expect over this distance (maybe you can use the 2018 spatial survey to estimate this)?

Response:

Thank you for raising this important question. To address it, we conducted a spatial analysis using the 2022-2024 hydrographic surveys and added a dedicated section 4.2 ("Spatial variability"). This analysis included five near-synoptic cruises with repeated sampling locations (August 2022, May and August 2023, May and August 2024) and incorporated data from Float 1 and 2 that overlapped with these cruises.

For PW properties, we found considerable spatial variability along isopycnals and at constant depths. The revised manuscript states in Lines 216-217, "PW seasonality is better assessed

from fixed-point observations at the Monitoring Station, since the pronounced spatial variability in PW would obscure the seasonal patterns if Float data are used".

In contrast, WGIW exhibits minimal spatial variability along isopycnals. Synoptic observations within the WGIW fall along a narrow line in the T-S space for all five cruises. Some spatial variability is present at constant depths, but it is relatively small and decreases with depth. In the revised manuscript, we provide a summary Table 2 for this analysis. Across the five cruises, PW/WGIW boundary depth varied with a standard deviation of 20 to 30m among all sampled locations. Density variability averaged approximately 0.023 kg m$^{-3}$ at 300m and 0.014 kg m$^{-3}$ at 400m depths, based on within-cruise variability in density at each depth. Accordingly, the revised manuscript states in Lines 212-214, "Because floats sampled the deeper parts of the bay, analysis of WGIW seasonality relies largely on float data. These observations can be used to assess WGIW variability along isopycnals with confidence, as well as at depth levels, provided that the variability in time exceeds the background spatial variability noted above".

We now address your question regarding Float 1 specifically. Its wide spatial coverage between August 2022 and May 2023 is considered acceptable for studying WGIW properties, because the temporal signals this Float captured with respect to WGIW renewal at 400m depth (0.05 kg m$^{-3}$ increase in November-December 2022 and 0.03 kg m$^{-3}$ increase in spring 2023) exceed the background spatial variability observed in the summer cruises of these years. Moreover, Float 1 moved very little during the spring renewal. As we noted in the methods (Lines 94-96), between the 6th of February and the 4th of April, 2023, the float was profiling underneath the sea ice, with no known positions. But between the acquisitions with known positions, the float's position changed only by 3.1km. During the autumn renewal in November-December 2022, Float 1 drifted by about 13km. Below is a figure showing positions covered during this time with black markers. Yet, over this time, the WGIW boundary shoaled by >100m, by far exceeding the standard deviation in boundary depth noted in spatial analysis. This demonstrates that the observed renewal reflected a robust signal rather than an artefact of float drift.

[Figure]

Changes in the manuscript:

- Conducted a spatial analysis using the 2022-2024 hydrographic surveys and added a new section 4.2 "Spatial variability", with new Fig.4 and Table 2.
- The answer to the reviewer's question is directly incorporated in Lines 212-217.

Line 319: I don't follow the statement that wind forcing was not evident during this period. It certainly looks like upwelling favourable winds were present at that time?

Response:

Thank you for pointing this out. The original statement about wind forcing not being evident during spring 2024 was unclear. In the revised manuscript, we present a more quantitative analysis of winds (added in response to the second reviewer's suggestion), which confirms that upwelling-favourable winds were indeed present during this period, but they were weaker than in spring 2023.

Changes in the manuscript: Updated Lines 300-306 with analysis of winter-spring 2024 conditions

Line 323-334: A lot of this paragraph would fit better into the discussion. There is some redundancy between the results and the discussion, which I think leads to some confusion and adding to the length of the paper.

Response: We moved this into discussion and revised the manuscript to reduce such redundancy throughout.

Changes in the manuscript:

Updated lines 307-312 to present the results of Fig. 8d and Fig.9 only, without interpretation. Incorporated the moved text into the discussion.

Line 333: I don't see 1.5C, maybe 1C increase?

Response: Thank you for catching this error. The value has been corrected from 1.5 °C to 1 °C.

Changes in the manuscript:

Corrected in revised Line 311.

Line 359: You cite 'typical' ranges here from previous work, but not anything about the range? That is, is it reasonable for this feature to be advective given variability in these conditions?

Response:

To clarify, our comparison was made against the minima observed at ~150 m in Davis Strait. Over the six years shown in Fig. 4 of Gladish et al. (2015, part 2), salinity at this depth rarely fell below 33.9 g kg$^{-1}$ (never below 33.8 g kg$^{-1}$), and density generally remained above $\sigma_0 = 27$ kg m$^{-3}$, expect during the anomalous year of 2011. In contrast, at the Monitoring Station in Disko Bay, autumn minima at 150 m were consistently lower : $\sigma_0 \approx 26.9$ kg m$^{-3}$ and $S_A \approx 33.84$ g kg$^{-1}$ in 2022; $\sigma_0 \approx 26.7$ kg m$^{-3}$ and $S_A \approx 33.59$ g kg$^{-1}$ in 2023; $\sigma_0 \approx 26.7$ kg m$^{-3}$ and $S_A \approx 33.58$ g kg$^{-1}$ in 2024. Because these values fall below the Davis Strait minima, we find that local freshwater sources within Disko Bay must also contribute, rather than the feature being fully explained by advection from the WGC.

Line 363: You cite both SGD and SMW are sources of freshwater, which is true. But they have one fundamental difference. For iceberg melt, the depth of entrainment is much much shallower than for SGD, so it's really the SGD that is controlling the depth of the GMW layer (well, and the sill). There is some new literature on refluxing at the icefjord sill (Hager et al. 2024) that might be useful as well to explore. This whole section is relatively long for describing a process that other studies have shown already. It's not really the novel or interesting part of this present study- that's more on the WGIW properties in my opinion.

Response:

Thank you for this constructive comment and suggestion on how to improve this section. In the revised manuscript, we streamlined and refocused the discussion of freshwater sources in Ilulissat Icefjord. We now outline the main factors that controlling the production and depth of GMW, treat iceberg melt separately by emphasising its role in modifying water mass properties within the fjord. In doing so, we refer to Hager et al. (2024) and other relevant studies. This section has been shortened to provide a concise summary of freshwater sources before shifting focus to the properties of its export into Disko Bay.

Changes in the manuscript: Former Lines 363-385 revised into Lines 338-354.

Line 393: do you mean 'width' here instead of height? As you also have draft? I'm not sure any of those papers actually have measurements of iceberg draft, but assume some geometry based on above water volume and ice/ocean densities.

Response:

We agree that our wording was unclear, and we have revised it accordingly. We now explicitly describe the assumed iceberg geometry: starting from the "small iceberg" with an area of ~1800 m$^2$ (Scheick et al., 2019), then representing such iceberg as 130m in length and width, from which we estimate it is 65m thick (freeboard+draft, following Enderlin et al., 2016 and their Figure 3a), and has a draft of 55m based on a freeboard-to-draft ratio of 1:7 (Cenedese and Straneo, 2023). We have also replaced "height" with "thickness" to avoid confusion.

Changes in the manuscript:

Lines 355-362 implement the above response.

Section 5.3: I am not sure this section says very much. A lot of it is speculation about how the results might be important to marine ecosystems. The last paragraph in particular is vague. Adding some meat/content would be good. For example, can you show the cyclonic circulation sense in the 2018 spatial survey (not just in T/S but in dynamic sections)?

Response:

We appreciate this comment and agree that section 5.3 was largely speculative and did not add substantial value. To maintain focus on the core results and avoid unnecessary speculation, we have removed this section.

Changes in the manuscript:

Section 5.3 has been removed.

Line 446: 'de-seasoning' is not a word.

This wording is removed with the deletion of section 5.3.

Line 453: This sentence and the one before it are confusing. They say nutrients are entrained deep in the icefjord, flow out into Disko Bay and then into the icefjord? Maybe I'm missing something here.

This wording is removed with the deletion of section 5.3.

General comment: Is it possible to update the schematic or create your own schematic (Fig 15 of Gladish et al (part 2))? This might help the reader understand what is new here or what is validating previous ideas (which is important too). I think the data here are very cool!

Response:

Thank you again for your positive and constructive suggestions. In the revised manuscript, we created an update to the schematic Figure 15 of Gladish et (2015). Our results provided observational evidence for the mechanism they propose in Spring/Summer (lower panel of their figure), which was well-represented in their schematic, and we retained it as is. We updated the upper panel to include the possible wind-driven renewal mechanism in autumn, which was not considered in the original schematic.

Changes in the manuscript: Added new Fig. 10 and reference to the figure in Lines 405-407, 412-413, 433-435.

**Response to Reviewer#2**

We sincerely thank you for your positive and constructive review of our manuscript. Your feedback helped us to improve the clarity and readability of the manuscript. We hope the addition of a new schematic figure and a more quantitative analysis of wind forcing, as you recommended, will assist readers in understanding the key mechanisms of WGIW renewal more clearly.

Please find our detailed responses in blue below. Most of your comments have been fully addressed. In addition, several further improvements were made while addressing Reviewer 1's comments, including incorporation of spatial surveys from 2022-2024, extension of Monitoring Station data until November 2024, and a more concise and structured presentation of our results and discussion. These changes have refined our interpretation of the source of the observed autumn along-isopycnal warming, whilst ensuring that the results on WGIW renewal remain central to the paper.

Best regards,

Linda Latuta on behalf of all co-authors.

**Reviewer#2 Summary**

This manuscript by Latuta et al. presents analyses of hydrographic dataset in Disko Bay, Greenland. It provides insights into the seasonality and spatial variability of the water masses in the bay system and gives explanations on the potential physical mechanisms that drive the seasonal cycles. Overall, I believe the manuscript would contribute to further our understanding of the hydrography in this area and the finding are significant and timely. But I also think the presentations could be improved and some clarifications should be made before acceptance.

**General comments**

1. A key explanation shown in this work is the identification of wind-driven WGIW renewal via upwelling-favorable conditions at the EDS. However, this mechanism (e.g., around line 305) is complex and may be difficult to visualize for some readers. I recommend including a schematic figure showing the seasonal wind changes, upwelling over EDS and the resultant basin renewal.

Response:

We appreciate this valuable suggestion. To improve clarity, we added a new schematic figure illustrating the seasonal mechanisms of WGIW renewal. The schematic builds on Fig. 15 of Gladish et al. (2015, part 2) but is adapted to highlight the processes identified in our study. In particular, it illustrates wind-driven upwelling over the Egedesminde Dyb sill and the resulting basin renewal observed under favourable wind forcing in autumn 2022. We believe this addition will help readers to visualise the dynamics better.

Changes in the manuscript: Added new Fig. 10 and reference to the figure in Lines 405-407, 412-413, 433-435.

2. The manuscript suggests that upwelling-favorable winds caused WGIW renewal in November and December 2022, but the upwelling effect was not as effective during autumn and winter 2023-2024 (50m uplift; line 317). It would be helpful to provide an estimate of the threshold of the wind stress or its duration needed to cause upwelling sufficient to lift WGIW over EDS. Is there any lag between wind stress and the dense water renewal? Could the authors quantify this relationship?

Response:

Thank you for this helpful suggestion. In the revised manuscript, we now quantify the relationship between wind forcing and calculated vertical velocities. Using hourly data, we define "strong upwelling" as $W_E \geq 0.45$ m day$^{-1}$ (upper quartile of the hourly data distribution), typically derived under $\tau_y \leq -0.06$ N m$^{-2}$. Throughout the revised results section 4.5, we implement a more quantitative representation of wind forcing during the periods, contrasting the conditions to the "strong upwelling" threshold and comparing the conditions between years.

Cross-correlation analysis shows that the Ekman response follows negative wind stress within hours, as expected given its derivation directly from the wind stress (Eq. 5). When relating upwelling to WGIW boundary depth, we found only a weak statistical relationship. This is expected for several reasons, and we note that quantifying this relationship with the available observations is complicated. First, our vertical velocities are derived from wind stress and represent the potential Ekman pumping over the EDS. However, we lack hydrographic observations west of or over EDS to observe the actual upwelling, determine the actual density of the uplifted waters, the magnitude of that vertical displacement, and the extent to which other processes, such as coastal upwelling, may have contributed. Second, the vertical uplift at the sill is not necessarily proportional to uplift within the Disko Bay basin below sill depth. Third, the WGIW boundary depth time series combines data from two Floats and Monitoring Station data. These represent different sampling locations, and the WGIW boundary depth itself varies by ~30m across Disko Bay (new Table 2). This inherent spatial variability limits our ability to define a robust quantitative relationship between wind forcing and WGIW boundary depth. Finally, the advective timescales between the inflow of dense waters across EDS and their arrival at the observation sites are not well constrained.

Changes in the manuscript:

Results section 4.5: Added quantitative representation of wind forcing during the outlined periods: Lines 280-286, 290-293, 296-299, 303-306.

Discussion: Lines 407-408, 410-416 implement the results and the above response to the reviewer.

3. The sampling frequency in Table 1 seems a bit long to capture variability in shorter time scales. It would be great if the authors could discuss whether the variability in higher frequencies would affect the water mass exchanges in the bay system.

Response:

We appreciate this comment and suggestion. While the sampling frequency of our data may appear sparse, it is unique in that it includes observations from late autumn, winter, and early spring — periods that are rarely sampled in Disko Bay. The Monitoring Station profiles (weekly-monthly, repeated three years, with a targeted autumn campaign in 2023 to enhance observational frequency), combined with the floats sampling at 5-day intervals, provide the longest and highest-frequency seasonal timeseries available. Together, these complementary datasets resolve the seasonal processes of interest, with signals that are persistent and repeated from year to year.

We agree that higher frequency variability is certainly present, but in the context of seasonal forcing, we believe the main physical processes are well captured. In the interest of concision (and in response to Reviewer 1), we have substantially shortened the manuscript and therefore, respectfully, choose not to add further discussion of higher-frequency variability, which we view a outside the scope of this study.

**Specific comments**

Figure 1 and lines 61-70: I notice the lack of labels for locations in the diagram, and it was difficult to follow the statement without further searching for another map online. Please include some key locations such as Vaigat Strait to help the reader navigate.

Changes in the manuscript:

Fig. 1 has been revised to incorporate the implemented suggestions.

Line 165 and Figure 2 caption: I realize the mixing line definition is not explicitly indicated in the Figure 2 caption. It would be more straightforward if the reader could find the definition of the mixing line end-points without referring back to the T-S statement in the manuscript.

Response: Thank you for this helpful suggestion.

Changes in the manuscript: Revised the caption of Fig.3 (former Fig.2), defining the mixing line end-points.

Figure 7a: please label the boxes with eastern, northern, and central.

Response: In the revised manuscript, the former Fig.7 (2018 spatial analysis) has been removed, as we now use 2022-2024 spatial surveys.

Changes in the manuscript: Ensured that all revised figures include clear labelling of data sources and regions/locations, e.g., marking towns that we refer to in-text in the new Fig. 4 based on 2022-2024 spatial surveys.

Line 319: it seems that in March and April 2024, the wind forcing and upwelling were strong (the same period in Figure 9c as the orange shading in Figure 9ef). Why does the statement here say "not evident"?

Response:

Thank you for pointing this out. The original statement about wind forcing not being evident during spring 2024 was unclear. Our revised analysis, strengthened with a more quantitative assessment of wind forcing (see also response to general comment #2), confirms that upwelling-favourable winds were present in spring 2024. However, they were weaker than those observed in spring 2023.

Changes in the manuscript:

Updated Lines 300-306 with clarified analysis of winter-spring 2024 conditions

Figure 10: please label the panels (a) and (b).

Changes in the manuscript:

Panels in Fig. 9 (formerly Fig. 10) are now labelled in the revised manuscript.